# Is Self-Supervised Contrastive Learning More Robust Than Supervised Learning?

Figure 1: We conduct a series of robustness tests based on data distribution corruptions from micro to macro levels, to study the behavior of contrastive and supervised learning beyond accuracy. Our results reveal that contrastive learning is usually more robust than supervised learning to *downstream* corruptions ($\Delta_{CL}^{D} < \Delta_{SL}^{D}$), while shows *opposite* behaviors to *pre-training pixel- and patch-level corruptions* ($\Delta_{CL}^{P} > \Delta_{SL}^{P}$) and *pre-training dataset-level* corruptions ($\Delta_{CL}^{P} < \Delta_{SL}^{P}$), where $\Delta$ is the accuracy drop from uncorrupted settings.

## Abstract

Prior work on self-supervised contrastive learning has primarily focused on evaluating the recognition accuracy, but has overlooked other *behavioral* aspects. In addition to accuracy, *distributional robustness* plays a critical role in the reliability of machine learning models. We design and conduct a series of robustness tests to quantify the behavioral differences between contrastive learning and supervised learning to downstream and pre-training data distribution changes. These tests leverage data corruptions at multiple levels, ranging from pixel-level distortion to patch-level shuffling and to dataset-level distribution shift, including both natural and unnatural corruptions. Our tests unveil intriguing robustness behaviors of contrastive and supervised learning: while we generally observe that contrastive learning is more robust than supervised learning under downstream corruptions, we surprisingly discover the robustness vulnerability of contrastive learning under pixel and patch level corruptions during pre-training. Furthermore, we observe the higher dependence of contrastive learning on spatial image coherence information during pre-training, e.g., it is particularly sensitive to global patch shuffling. We explain these results by connecting to feature space uniformity and data augmentation. Our analysis has implications in improving the downstream robustness of supervised learning, and calls for more studies on understanding contrastive learning.

## 1 Introduction

In recent years, self-supervised contrastive learning (CL) has demonstrated tremendous potential in learning generalizable representations from unlabeled datasets (Chen et al., 2020b; He et al., 2020; Grill et al., 2020; Caron et al., 2020; Chen & He, 2021; Zhong et al., 2021b). Current state-of-the-art CL algorithms learn representations from ImageNet (Deng et al., 2009) that match or even exceed the accuracy of their supervised learning (SL) counterparts on ImageNet and downstream tasks.

However, beyond accuracy, little attention is paid on comparing other *behavioral differences* between contrastive learning and supervised learning, and even less work investigates the robustness during pre-training. Robustness is an important aspect to evaluate machine learning algorithms. For example, robustness to long-tail or noisy training data allows the learning algorithm to work well in a wide variety of imperfect real-world scenarios (Wang et al., 2017). Robustness of the model output across training iterations enables anytime early-stop (Hu et al., 2019) and smoother continual

learning (Shen et al., 2020). Robustness to input corruptions at test-time plays an important role in reliable deployment of trained models in safety-critical applications, as signified by the existence of adversarial examples (Goodfellow et al., 2015; Salman et al., 2020) and the negative impact of domain shift (Zhao et al., 2019).

In this paper, we investigate whether CL and SL behave robustly to data distribution changes. In particular, how does changes in *data* affect behaviors of algorithms? And do CL and SL behave similarly? To this end, we design a wide-spectrum of corruptions as shown in Figure 1 to alter data distribution and conduct comprehensive experiments, with different backbones, CL algorithms and datasets. The corruptions are carefully selected to be multi-level, targeting both human-recognizable and unrecognizable structural information, and are rooted in prior literature: pixel-level corruptions distorts intensity distribution, patch-level shuffle corrupts spatial structure (Ge et al., 2021; Neyshabur et al., 2020; Zhang et al., 2017; Hendrycks & Dietterich, 2019), and dataset-level class imbalance (Liu et al., 2022; 2019; Samuel & Chechik, 2021) and GAN (generative adversarial network) synthesis (Jahanian et al., 2021) shift the overall distribution.

Our main results consist of two sets of experiments: The first set investigates the *downstream* robustness of pre-trained models towards corruptions of downstream data. The second set studies the robustness under *pre-training* data corruptions – when the accuracy degradation of an algorithm to some corruption is large, it suggests that the algorithm may leverage such information as learning signal. Note that our work is inspired by Zhang et al. (2017) and Ribeiro et al. (2020) and follows a similar *empirical exploratory analysis*, rather than a regular adversarial robustness paradigm.

We deliver a set of **intriguing new discoveries**. We generally observe that CL is consistently more robust than SL to *downstream* corruptions. Meanwhile, contrastive learning on corrupted *pre-training* leads to *diverging* observations: CL is more robust to dataset-level corruption than SL, but much less so to pixel- and patch-level corruptions. Moreover, we discover the higher dependence of contrastive learning on spatial information during pre-training, such that a global patch shuffling corruption harms feature learning greatly.

To understand why pre-trained CL models are more robust to downstream corruptions, we analyze the learning dynamics through feature space metrics and find that CL yields larger *overall* and steadily-increasing *per-class feature uniformity* and higher stability than SL. The instance-level CL objective might capture richer sets of features not limited to semantic classes. Therefore, the per-class uniformity or intra-class variation is not compressed as hard as in SL. This allows the CL models to generalize to unseen corrupted downstream data better than SL. Such hypothesis aligns well with several recent attempts to understand CL (Zhao et al., 2021; Chen et al., 2021a; Liu et al., 2022). An immediate consequence of our insight is an improvement to supervised pre-training by adding a uniformity regularization term to explicitly promote intra-class variance, where the test-time data corruption robustness is improved.

As for CL's vulnerability to pre-training data corruptions such as patch shuffling, we speculate that CL is more dependent on the spatial structure of images, and the introduction of high-frequency noise undermines the long-scale spatial coherence of natural images. For example, with global patch shuffling, the random resized cropping used in CL is no longer a proper data augmentation. We verify our intuition by manipulating data pre-processing and analyzing attention maps. We find that corrupting after standard data augmentation recovers a substantial amount of robustness, making CL comparably robust to SL.

We summarize **our contributions** as follows. (1) We design extensive distributional robustness tests to study the behavioral differences of CL and SL systematically. (2) We discover diverging robustness behaviors between CL and SL, and even among different CL algorithms. (3) We offer analyses and explanations for such observations, and show a simple way to improve the downstream robustness of supervised learning. We claim our paper as an empirical study. We hope our findings can serve as an initial step to fully understand CL's behaviors beyond accuracy and inspire more future studies to explore such aspects through theoretical analysis.

## 2 RELATED WORK

**Self-Supervised Learning (SSL) and Contrastive Learning (CL).** Remarkable progress has been made in self-supervised representation learning from unlabeled datasets (Chen et al., 2020b; He

et al., 2020; Grill et al., 2020; Caron et al., 2020; Chen & He, 2021). This paper focuses on a particular kind of SSL algorithm, contrastive learning, that learns augmentation invariance with a Siamese network. To prevent trivial solution, contrastive learning pushes negative examples apart (MoCo (He et al., 2020; Chen et al., 2020d; 2021b), SimCLR (Chen et al., 2020b;c)), makes use of stop-gradient operation or asymmetric predictor without using negatives (BYOL (Grill et al., 2020), SimSiam (Chen & He, 2021), DINO (Caron et al., 2021)), or leverages redundancy reduction (BarlowTwins (Zbontar et al., 2021)) and clustering (DeepCluster-v2 and SwAV (Caron et al., 2020)). In addition to augmentation invariance, generative pre-training (Ramesh et al., 2021; Bao et al., 2022; He et al., 2022) and visual-language pre-training (Radford et al., 2021) are promising ways to learn transferable representations.

There is a growing body of literature on understanding SSL. Wang & Liu (2021) decomposes the contrastive objective into alignment (between augmentations) and uniformity (across entire feature space) terms. Uniformity can be thought of as an estimate of the feature entropy, which we use to study the feature space dynamics during training. Wang & Isola (2020) makes connection between uniformity and the temperature parameter in contrastive loss, and finds that a good temperature can balance uniformity and tolerance of semantically similar examples. Zhao et al. (2021) discovers that SSL transferring better than SL can be due to better low- and mid-level features, and the intra-class invariance objective in SL weakens transferability by causing more pre-training and downstream task misalignment. Ericsson et al. (2021) studies the downstream task accuracy of a variety of pre-trained models and finds that SSL outperforms SL on many tasks. Cole et al. (2022) investigates the impact of pre-training data size, domain quality, and task granularity on downstream performance. Chen et al. (2021a) identifies three intriguing properties of CL: a generalized version of the loss, learning with the presence of multiple objects, and feature suppression induced by competing augmentations. Our work falls into the same line of research that attempts to understand SSL better. However, we investigate from the angle of *robustness behavior comparison* between SSL/CL and SL.

**Robustness and Data Corruption.** The success of learning algorithms is often measured by some form of task accuracy, such as the top-1 accuracy for image classification (Deng et al., 2009; Krizhevsky et al., 2009; Coates et al., 2011; Wah et al., 2011), or the mean average precision for object detection (He et al., 2020; Zhong et al., 2021a; 2020). Beyond accuracy, robustness is another important measure (Hendrycks & Dietterich, 2019), and there are benchmarks and metrics proposed for SL (Nado et al., 2021). Robustness is becoming more studied for SSL. Chuang et al. (2022) tries to improve CL's robustness to noisy positive views. Goyal et al. (2022) reveals that vision models are more robust and fair when pre-trained on uncurated images without supervision . Albuquerque et al. (2020) leverages SSL's robustness to out-of-domain examples to facilitate domain generalization in multi-task learning . We use "robustness" to refer to the ability of learning algorithms to cope with systematic train or test data corruptions. Under the supervised setting, deep models are shown to train successfully (albeit not to generalize) under pixel shuffling corruption and random labels, even though they are not human-recognizable anymore (Zhang et al., 2017).

Adversarial robustness (Szegedy et al., 2014; Goodfellow et al., 2015; Madry et al., 2018; Shafahi et al., 2019; Chen et al., 2020a) is a related but different concept, which refers to the model's ability to defend against adversarial attacks. An adversarial attack (Szegedy et al., 2014; Goodfellow et al., 2015) is a perceptually indistinguishable perturbation to a *single* image that fools the model. Adversarial training (Madry et al., 2018; Shafahi et al., 2019) is a technique to achieve adversarial robustness. Self-supervised perturbation is explored in adversarial attack and training (Naseer et al., 2020; Kim et al., 2020). Hendrycks et al. (2019) shows that SSL models possess better adversarial robustness. Fan et al. (2021) improves the adversarial robustness transferability of CL. Our definition of robustness differs from adversarial robustness – we use robustness to analyze the tolerance of learning methods to *systematic data corruptions* (rather than per-image imperceptible perturbation).

There are many types of data corruptions in prior work. The most common data corruptions, such as random resizing and cropping, flipping, and color jittering, appear as data augmentation in SL and SSL (He et al., 2016; 2020; Chen et al., 2020b). The learned representation is encouraged to be invariant to such corruptions. Hendrycks & Dietterich (2019) proposes a set of corruptions complementary to ours. Block shuffling (our image global shuffling) has been used to study what is transferred in transfer learning (Neyshabur et al., 2020) and as negative views with diminished semantics in contrastive learning (Ge et al., 2021). Cole et al. (2022) tampers data quality in SimCLR and SL training by salt-and-pepper noise, JPEG, resizing, and downsampling, and tests on clean data. We use a broader set of data corruptions and test on the corrupted data as well. A recent

work (Jahanian et al., 2021) studies generative models as an alternative data source for contrastive learning. They focus on comparison with real data, while we emphasize the behavior difference of SSL and SL in response to the generative data source. Feature backward-compatibility (Shen et al., 2020) is related to our stability analysis of feature dynamics. Recently, Goyal et al. (2021a) studies the effectiveness of SSL on uncurated class-imbalanced data. Liu et al. (2022) also notices that SSL tends to be more robust to class imbalance than SL. We bring extra insights over them. We consider *both pre-training and downstream* robustness and compare *CL vs. SL behaviors*, while Goyal et al. (2021a) only focuses on downstream and compares dataset scale. Our investigation suggests that pre-train behavior can be *opposite* to downstream. Liu et al. (2022) only studies class imbalance, but we also consider image structural corruptions.

## 3 METHOD

We define distributional robustness as robustness against various distribution shifts of input images by carefully-designed data corruptions, and evaluate the distributional robustness of different algorithms by observing the impact. We refer to the behavior of a learning algorithm as how it learns the representations and how such learning evolves throughout training. To what extent will such corruptions influence the performance? Will there be consistent trends that depend on the type of the corruptions? And will there be a behavioral difference between CL and SL?

### 3.1 ROBUSTNESS TESTS

The common way of using CL or SL models is through the pre-training and fine-tuning paradigm (Chen et al., 2020b; He et al., 2020; Zhong et al., 2021a;b). A neural backbone is pre-trained on a large-scale dataset such as ImageNet (Deng et al., 2009) or composite dataset of images scrapped from Internet with mixed quality (Radford et al., 2021), and transferred to initialize downstream models or inference. Therefore, it is crucial to consider the impact of data corruptions in both the pre-training and the downstream phases. Since data corruption destroys certain information by design, both settings on the corrupted data are expected to yield degraded performance. Specifically, we perform the following two complementary types of tests.

**Robustness Test I: Downstream data corruption.** In this test, the pre-training algorithm is run on the clean version of the pre-training dataset. For a given downstream dataset, we evaluate the pre-trained model's accuracy on its original version and various corrupted versions. This assesses the robustness of the algorithm by looking at the pre-trained model's robustness behaviors.

**Robustness Test II: Pre-training data corruption.** To assess the algorithm's robustness to pre-training data corruptions, we run the pre-training algorithm on the corrupted version of the dataset, and then evaluate the final model's accuracy on either the corrupted test set or the original test set. The test set can be in-domain (the same domain as the train set) or out-domain (a different domain from the train set).

**Robustness Metric.** In both cases, the robustness is measured by the degradation in accuracy caused by certain data corruption. An algorithm is more robust if the degradation is smaller. Denote $\mathcal{D}_{\text{original}}$ as the original dataset and $\mathcal{D}_{\text{corrupted}}$ as the corrupted dataset. For an algorithm $\text{Alg} \in \{\text{CL}, \text{SL}\}$, we define $\Delta(\text{Alg})$ as $\frac{\text{Acc}(\text{Alg}, \mathcal{D}_{\text{original}}) - \text{Acc}(\text{Alg}, \mathcal{D}_{\text{corrupted}})}{\text{Acc}(\text{Alg}, \mathcal{D}_{\text{original}})}$. The essential question we are asking is whether $\Delta(\text{CL})$ is consistently larger or smaller than $\Delta(\text{SL})$ across different data corruptions.

We use two methods to obtain the test accuracy in the above equation. The first is **linear evaluation**, where we train a linear classifier on top of the learned representations on the train split and evaluate on the test split. The second is **KNN evaluation** following Wu et al. (2018), where the prediction is the exponential-distance weighted average of the $K$ nearest neighbors in the train split of any test data point, measured by the normalized feature vectors. The KNN evaluation effectively leverages an non-parametric classifier, therefore no classifier training is required.

### 3.2 DATA CORRUPTION TYPES

There is a natural hierarchy of data corruptions ranging conceptually from *micro-level* to *macro-level*. We describe our choices below (also illustrated in Figure 1). Note that our data corruption is

different from data augmentation randomly applies transform on a per-image basis. In our case, a fixed random transformation (e.g., the $\gamma$ in gamma distortion or the permutation order in shuffling) is decided first and then applied consistently across all images. We effectively transform the entire dataset with the corruption method.

We emphasize that our purpose is not only to study human-recognizable distortions, but to evaluate pre-training algorithms' behavior under various distortions. To this end, our collection of corruptions is designed to be representative and comprehensive: while some of them are practical (natural corruptions, imbalance), others are purposefully introduced to distort certain structural information (shuffling). A similar flavor of behavior study was seen in Zhang et al. (2017).

**Pixel-Level Corruption.** The pixel intensity distribution is altered, but neither the spatial layout of each image nor the overall data distribution is changed. Here, we deliberately pick gamma distortion and selected ImageNet-C corruptions (Hendrycks & Dietterich, 2019) since they are not part of the conventional data augmentation pipeline.

- **Gamma distortion:** Gamma distortion remaps each RGB pixel intensity ($\in [0, 255]$) according to $x \rightarrow \lfloor 255 \times (x/255)^{\gamma} \rfloor$, where $\gamma > 0$ is a tunable parameter. Larger or smaller $\gamma$ shifts the intensities darker or brighter, respectively. Due to quantization error, there will be part of the intensity information lost during the process.
- **ImageNet-C:** ImageNet-C (Hendrycks & Dietterich, 2019) focuses on natural, human-recognizable corruptions such as noises, blurring, weathers, etc. We pick shot noise, defocus blur, and JPEG compression in our pre-training robustness experiments.

**Patch-Level Corruption.** Inspired by Zhang et al. (2017), we consider random shuffling. Note that patch shuffling is not commonly used in the standard augmentation pipeline. We are curious about what behaviors CL and SL will exhibit when patch shuffling destroys certain structural coherence.

- **Global shuffling:** We break down the image into patches and shuffles the patches according to a fixed random order. Specifically, given the image size $s \times s$ and the patch size $p \times p$, the image is divided into $s/p \times s/p$ patches. Global shuffling destroys the global spatial structure of an image but preserves the local structure. The image becomes less structured with a smaller patch size.
- **Local shuffling:** Inversely to global shuffling, local shuffling randomly permutes the pixels inside each local $p \times p$ patch by a fixed random order, but keeps the global ordering of patches. It damages the local image structure while preserving the overall global structure. The image becomes less structured with a larger patch size.

**Dataset-Level Corruption.** We hereby consider corruptions happening at the whole dataset distribution level, as the previous two corruptions only change the images but not the overall distribution.

- **Synthesized data:** Synthesized data is popularizing (e.g., DALL·E 2 (Ramesh et al., 2022)) and studied to replace real data (Jahanian et al., 2021). We utilize GAN (Karras et al., 2020) to generate a synthesized dataset $\mathcal{D}_{GAN}$ and replace $\mathcal{D}_{original}$. We then measure and compare $\Delta(Alg)$ between these two datasets. Oftentimes, the generated distribution is not perfectly aligned with the real distribution, therefore training with the generative data source may lead to degradation in accuracy of clean data or downstream performance.
- **Class imbalance:** Real-world data often follows a long-tail distribution, where a few common semantic classes have lots of examples while many tail classes have few examples (Kang et al., 2020; Samuel & Chechik, 2021). However, benchmark datasets such as CIFAR and ImageNet are curated and class-balanced. We consider the widely-used variant of ImageNet, ImageNet-LT (long-tail) (Liu et al., 2019), with maximally 1280 images and minimally 5 images per class. For comparison, we construct ImageNet-UF (uniform), a class-balanced subset of ImageNet which contains the same number of images as ImageNet-LT (115K). We test whether moving from pre-training on ImageNet-UF to ImageNet-LT would lead to different behaviors between CL and SL.

### 3.3 EXPERIMENT SETUP

**Datasets.** We pre-train on CIFAR-10 (Krizhevsky et al., 2009), ImageNet (Deng et al., 2009) and its variants to evaluate the distributional robustness of CL and SL to pre-training data corruptions. We use CIFAR-10/100 (Krizhevsky et al., 2009), STL-10 (Coates et al., 2011), and fine-grained classification datasets (Cars (Krause et al., 2013) and Aircrafts (Maji et al., 2013)) to analyze the performance of the pre-trained models on the corrupted downstream tasks. For fair comparisons, we use the same data augmentation across methods when we need to train any model.

Table 1: Robustness Test I: *downstream pixel- and patch-level* corruptions with *ResNet-50* backbone. Models pre-trained on original ImageNet are downloaded from corresponding official websites ('IN Acc:' reference ImageNet Val accuracy). We consider 5 downstream datasets. For each dataset, we report the averages of 6 corruption settings: gamma distortion $\gamma = \{0.2, 5\}$, global and local shuffling ($p = \{4, \text{image\_size}/4\}$). The image size is 32 for C-10/100, 96 for STL-10, and 256 for the rest; the corrupted images are resized to 224 as input to the network. We compute the KNN accuracy (K=50 for C-10/100 and STL-10, K=5 for others) on corrupted test sets and report $\Delta$ relative to the uncorrupted versions. Avg $\Delta$ is the average over the 5 datasets (darker shades indicate higher drops). This table only shows $\Delta$. Please refer to Appendix B.4 for more detailed accuracies. Contrastive learning models generally show lower accuracy drops and therefore higher downstream robustness than supervised models.

| Pre-train Alg | IN Acc | C-10 | C-100 | STL-10 | Car-196 | Air-70 | Avg $\Delta \downarrow$ |
|---|---|---|---|---|---|---|---|
| Sup | 76.1 | 31.5% | 45.3% | 31.0% | 51.2% | 39.9% | 39.8% |
| BYOL | 72.3 | 29.3% | 43.0% | 29.0% | 42.9% | 33.8% | 35.6% |
| SimSiam | 68.3 | 27.8% | 40.8% | 29.3% | 41.5% | 32.6% | 34.4% |
| MoCo-v2 | 71.1 | 31.3% | 45.2% | 31.0% | 39.7% | 31.3% | 35.7% |
| SimCLR-v2 | 71.0 | 31.5% | 45.4% | 30.8% | 43.0% | 31.7% | 36.5% |
| BarlowTwins | 73.5 | 26.7% | 39.8% | 29.7% | 43.0% | 34.4% | 34.7% |
| DeepCluster-v2 | 75.2 | 28.2% | 41.1% | 28.5% | 43.2% | 38.9% | 36.0% |
| SwAV | 74.9 | 26.8% | 39.3% | 28.6% | 41.4% | 36.3% | 34.5% |

Table 2: Robustness Test I: *downstream pixel- and patch-level* corruptions with *ViT* backbone. We show KNN accuracies and the $\Delta$'s on three datasets. Similar to Table 1, ViT CL models are also more robust than the two SL models, especially to gamma distortion. The generative method, MAE (He et al., 2022), is slightly more robust than CL to patch shuffling on CIFAR, but inferior on STL10 and more vulnerable to gamma distortion. We include average $\Delta$ for each algorithm across all datasets for a clearer comparison.

| Alg | Dataset | Orig | $\gamma$0.2 | $\gamma$2.5 | G4x4 | G24x24 | L4x4 | L24x24 | Avg $\Delta$ |
|---|---|---|---|---|---|---|---|---|---|
| ViT (Sup) | STL10 | 98.85 | 91.71 (7.2%) | 91.39 (7.5%) | 88.96 (10.0%) | 43.69 (55.8%) | 45.95 (53.5%) | 70.89 (28.3%) | 27.1% |
| Alg Avg $\Delta$: | CIFAR10 | 94.23 | 71.42 (24.2%) | 82.37 (12.6%) | 64.09 (32.0%) | 52.58 (44.2%) | 52.54 (44.2%) | 59.63 (36.7%) | 32.3% |
| 36.1% | CIFAR100 | 79.86 | 48.70 (39.0%) | 60.87 (23.8%) | 40.95 (48.7%) | 29.84 (62.6%) | 28.91 (63.8%) | 35.31 (55.8%) | 49.0% |
| DeiT (Sup) | STL10 | 98.64 | 97.58 (1.1%) | 98.01 (0.6%) | 92.92 (5.8%) | 46.99 (52.4%) | 45.60 (53.8%) | 73.22 (25.8%) | 23.3% |
| Alg Avg $\Delta$: | CIFAR10 | 95.37 | 90.66 (4.9%) | 92.78 (2.7%) | 73.24 (23.2%) | 59.48 (37.6%) | 53.10 (44.3%) | 59.65 (37.5%) | 25.0% |
| 28.7% | CIFAR100 | 78.23 | 68.98 (11.8%) | 73.00 (6.7%) | 49.86 (36.3%) | 34.81 (55.5%) | 29.49 (62.3%) | 36.12 (53.8%) | 37.7% |
| DINO | STL10 | 98.91 | 98.31 (0.6%) | 98.17 (0.7%) | 95.30 (3.7%) | 50.36 (49.1%) | 52.35 (47.1%) | 79.96 (19.2%) | 20.1% |
| Alg Avg $\Delta$: | CIFAR10 | 96.68 | 92.85 (4.0%) | 94.65 (2.1%) | 77.99 (19.3%) | 64.63 (33.2%) | 60.79 (37.1%) | 68.04 (29.6%) | 20.9% |
| 24.9% | CIFAR100 | 83.88 | 75.76 (9.7%) | 79.21 (5.6%) | 56.81 (32.3%) | 40.80 (51.4%) | 36.62 (56.3%) | 44.82 (46.6%) | 33.7% |
| MoCo-v3 | STL10 | 97.89 | 97.11 (0.8%) | 96.75 (1.2%) | 91.24 (6.8%) | 48.86 (50.1%) | 47.70 (51.3%) | 74.88 (23.5%) | 22.3% |
| Alg Avg $\Delta$ | CIFAR10 | 96.16 | 91.90 (4.4%) | 94.17 (2.1%) | 75.30 (21.7%) | 61.14 (36.4%) | 57.60 (40.1%) | 64.43 (33.0%) | 22.9% |
| 27.1% | CIFAR100 | 82.32 | 73.25 (11.0%) | 77.42 (6.0%) | 53.07 (35.5%) | 37.75 (54.1%) | 33.00 (59.9%) | 40.79 (50.4%) | 36.2% |
| MAE | STL10 | 90.74 | 83.54 (7.9%) | 87.42 (3.7%) | 72.54 (20.1%) | 46.35 (48.9%) | 46.20 (49.1%) | 60.15 (33.7%) | 27.2% |
| Alg Avg $\Delta$ | CIFAR10 | 77.06 | 71.00 (7.9%) | 72.04 (6.5%) | 61.25 (20.5%) | 55.06 (28.5%) | 53.31 (30.8%) | 56.99 (26.0%) | 20.0% |
| 25.4% | CIFAR100 | 53.70 | 47.72 (11.1%) | 49.46 (7.9%) | 37.18 (30.8%) | 30.93 (42.4%) | 29.36 (45.3%) | 34.18 (36.4%) | 29.0% |

**Models and Algorithms.** We benchmark a variety of self-supervised contrastive learning algorithms. These methods are carefully sampled to be representative. They include contrastive learning with negatives: SimCLR-v2 (Chen et al., 2020b;c), MoCo-v2 (He et al., 2020; Chen et al., 2020d); without negatives: SimSiam (Chen & He, 2021), the momentum based, BYOL (Grill et al., 2020); with redundancy reduction: BarlowTwins (Zbontar et al., 2021); and with clustering assignments: DeepCluster-v2 (Caron et al., 2020), SwAV (Caron et al., 2020). We test both CNN (standard ResNet-18/50 (He et al., 2016)) and Vision Transformer (ViT) (Dosovitskiy et al., 2021) backbones. For transformers, we leverage pre-trained models on ImageNet (Deng et al., 2009) from ViT (Dosovitskiy et al., 2021), DeiT (Touvron et al., 2021), DINO (Caron et al., 2021), MoCo-v3 (Chen et al., 2021b), and MAE (He et al., 2022) (which makes an interesting comparison as it is based on reconstruction rather than contrasting).

# 4 RESULTS

## 4.1 CL IS MORE ROBUST TO DOWNSTREAM DATA CORRUPTIONS THAN SL

We show the results of downstream robustness tests on various datasets with frozen ResNet-50 (He et al., 2016) in Table 1 and ViT (Dosovitskiy et al., 2021) in Table 2. Model checkpoints are obtained from VISSL (Goyal et al., 2021b) and the official code bases. They are pre-trained on the clean version of ImageNet. We employ pixel-level and patch-level corruptions and report KNN accuracy. The raw accuracy numbers are in Appendix B.4.

For pixel-level corruption, we pick gamma distortion with $\gamma = \{0.2, 5\}$. For patch-level corruption, we choose a small patch size and a large patch size for local and global shuffling each. In both tables, CL methods have demonstrated higher robustness (lower average $\Delta$) than SL. The same observation holds if we unfreeze the backbone and fine-tune fully with an additional linear layer as shown in Appendix B.5. Interestingly, not all CL methods are equally robust; even within the same method, models trained with different hyper-parameters (such as epochs) exhibit different levels of

robustness (Appendix B.4, B.5). With ResNet-50, we notice SimSiam, SwAV, and BarlowTwins to behave slightly more robust than others.

Table 3: Robustness Test II: *pre-training pixel- and patch-level* corruptions of CIFAR10 with ResNet18, and full ImageNet with ResNet50. We use linear evaluation. We discover that SL is more robust than CL in this scenario. While CL methods obtain average $\Delta$ about $20\%$, SL achieves $16.7\%$ for CIFAR10 and $7.9\%$ for ImageNet, which is lower than the best CL methods here (MoCo v2 and BYOL).

| | C10 orig | $\gamma$0.2 | G4x4 | G8x8 | L4x4 | L8x8 | Avg $\Delta\downarrow$ | IN orig | $\gamma$0.2 | G4x4 | L64x64 | Avg $\Delta\downarrow$ |
|---|---|---|---|---|---|---|---|---|---|---|---|---|
| Sup | 89.53 | 87.36 (2.4%) | 76.06 (15.0%) | 65.88 (26.4%) | 65.94 (26.3%) | 77.49 (13.4%) | 16.7% | 71.79 | 69.59 (3.1%) | 62.59 (12.8%) | 66.24 (7.7%) | 7.9% |
| MoCo-v2 | 88.73 | 85.84 (3.3%) | 67.18 (24.3%) | 60.51 (31.8%) | 63.35 (28.6%) | 76.90 (13.3%) | 20.3% | 64.06 | 61.07 (4.7%) | 35.02 (45.3%) | 57.63 (10.0%) | 20.0% |
| BYOL | 88.39 | 82.72 (6.4%) | 67.47 (23.7%) | 60.63 (31.4%) | 62.64 (29.1%) | 75.15 (15.0%) | 21.1% | 64.19 | 62.67 (2.4%) | 35.00 (45.5%) | 58.72 (8.5%) | 18.8% |
| Barlow | 88.89 | 80.49 (9.4%) | 68.34 (23.1%) | 61.13 (31.2%) | 62.53 (29.7%) | 75.28 (15.3%) | 21.7% | | | N/A | | |
| DINO | 84.75 | 69.27 (18.3%) | 64.26 (24.2%) | 55.83 (34.1%) | 58.57 (30.9%) | 68.96 (18.6%) | 25.2% | | | N/A | | |

Table 4: Robustness Test II: *pre-training pixel-level natural* corruptions of CIFAR10 with ResNet18 backbone, and full ImageNet (Deng et al., 2009) with ResNet50 backbone following (Hendrycks & Dietterich, 2019). We select MoCo-v2 (Chen et al., 2020d) to compare with SL on linear evaluation, and we pick shot noise, defocus blur, and JPEG compression as natural corruptions. On both datasets, SL achieves lower average $\Delta$, which aligns with unnatural corruptions during pre-training.

| | CIFAR10 Orig | Shot | Defocus | JPEG | Avg $\Delta\downarrow$ | ImageNet Orig | Shot | Defocus | JPEG | Avg $\Delta\downarrow$ |
|---|---|---|---|---|---|---|---|---|---|---|
| Sup | 89.53 | 88.04 | 90.9 | 87.08 | 0.9% | 71.79 | 69.34 | 66.89 | 70.44 | 4.0% |
| MoCo-v2 | 88.73 | 82.02 | 88.08 | 78.59 | 6.5% | 64.06 | 52.99 | 54.76 | 55.73 | 14.9% |

## 4.2 CL IS LESS ROBUST TO PRE-TRAINING PIXEL-LEVEL AND PATCH-LEVEL CORRUPTIONS

Contrary to downstream corruptions where CL demonstrates consistent higher robustness, whether CL is more robust than SL depends on the type of corruption during pre-training. Extensive experiments show that SL is more robust to pixel- and patch-level corruptions.

Table 3 shows the impacts of gamma distortion and patch shuffling on CL and SL during pre-training. We train SL for 30 epochs and CL for 200 epochs (except for DINO which is trained for 600 epochs) for comparable clean data accuracy via linear evaluation. The $\Delta$ of SL due to gamma distortion is $2.4\%$ which outperforms all the tested CL methods. For pre-training patch shuffling corruption, all CL methods behave similarly and less robustly than SL, except for the L8x8 case where Sup and MoCo-v2 are comparable. We also extend to natural pixel-level corruptions for MoCo-v2 and SL as shown in Table 4. While SL achieves $0.9\%$ and $4.0\%$ average $\Delta$ on CIFAR10 and full ImageNet respectively, MoCo-v2 obtains worse average $\Delta (6.5\%, 14.9\%)$. This aligns with previous observations from gamma distortion and shuffle. Additional experiments involving ViT backbone, longer training, ImageNet-100 are included in Appendix B.1, B.2 and B.3, which cover wider corruption settings and all report the same observations.

## 4.3 CL IS MORE ROBUST TO PRE-TRAINING DATASET-LEVEL CORRUPTIONS

To investigate pre-training distribution shift caused by synthesized data, we adopt a class-conditional StyleGAN2-ADA (Karras et al., 2020) trained on CIFAR-10 to generate a synthesize copy of same size. We train MoCo-v2 for 200 epochs and SL for 50 epochs (both ResNet-18 backbones) with different train/test data settings, reporting performance differences in Table 5. When training on the synthesized data and testing on the original CIFAR-10, MoCo-v2 only has $2.58\%\Delta$, greatly outperforming the supervised method with $8.44\%\Delta$. Evaluating on a GAN-synthesized test set yields similar observation – MoCo-v2 shows almost no drop while Sup drops $6\%$. Testing on out-domain CIFAR-100 delivers the same behavior.

Table 6 shows the impact of class imbalance. We use ImageNet-LT (long-tail) dataset to simulate the real-world long-tail class distribution (Liu et al., 2019), and we sample a balanced subset of ImageNet named ImageNet-UF (uniform), with the same size as ImageNet-LT. We train with ResNet-50 backbone and compare the recognition accuracy on the ImageNet-LT validation split of the fine-tuned linear classifiers on ImageNet-UF. Despite a gap between the baseline top-1 accuracy of MoCo-v2 and SL, we observe that the decline of MoCo resulting from pre-training on the long-tail rather than the uniform version is much smaller than SL. In fact, the MoCo performance appears to be insensitive to class balance or imbalance (the top-1 $\Delta$ is only $0.71\%$). This is contrary to SL, which shows a larger drop. The difference is more salient by looking at the low-shot ($< 20$ images per class), medium-shot, and many-shot ($> 100$ images per class) accuracy separately. Supervised pre-training on the long-tail version sacrifices the low-shot accuracy for a higher many-shot accuracy, whereas MoCo-v2 pre-training shows insignificant difference among the shots. Our observation is consistent with a contemporary work (Liu et al., 2022).

Table 5: Robustness Test II: *pre-training synthesized data*. C10/C100 refer to CIFAR-10/100. Interestingly, at absolute scale, MoCo shows higher downstream transfer accuracy to CIFAR-100 than SL, even through the 10 pre-training classes are only a small subset of the CIFAR-100. In all three evaluation settings, MoCo-v2 demonstrates much more robustness (on average, $\Delta_{\text{MoCo}} = 0.93\%$) than Sup ($\Delta_{\text{Sup}} = 7.71\%$) to the distribution shift of synthesized data.

| Alg | Data | C10 Test | C10 GAN Test | C100 Test | Avg $\Delta \downarrow$ |
|---|---|---|---|---|---|
| Sup | Orig C10 | 87.8 | 88.3 | 16.08 | - |
| | GAN C10 | 80.0 (8.88%) | 82.8 (6.23%) | 14.79 (8.02%) | 7.71% |
| MoCo-v2 | Orig C10 | 82.6 | 85.1 | 45.47 | - |
| | GAN C10 | 82.2 (0.48%) | 85.4 (-0.35%) | 44.27 (2.64%) | 0.93% |

Table 6: Robustness Test II: *pre-training class imbalance*. We compare MoCo and SL on ImageNet-LT (long-tail) (Liu et al., 2019) and ImageNet-UF (uniform). We train a linear classifier on ImageNet-UF and report accuraccies on ImageNet-LT-Val (20K images). Low-shot refers to classes with less than 20 images, many-shot with more than 100, and med-shot in between. MoCo shows less sensitivity to pre-train data imbalance than Sup with smaller $\Delta$ and variance.

| Alg | Data | Top-1 | Low | Med | Many |
|---|---|---|---|---|---|
| Sup | ImageNet-UF | 46.37 | 44.85 | 45.88 | 47.52 |
| | ImageNet-LT | 44.90 (3.17%) | 40.99 (8.61%) | 43.48 (5.23%) | 48.05 (-1.12%) |
| MoCo-v2 | ImageNet-UF | 32.36 | 30.63 | 31.66 | 33.84 |
| | ImageNet-LT | 32.13 (0.71%) | 30.99 (-1.18%) | 31.45 (0.66%) | 33.36 (1.42%) |

**Discussion.** We try to balance diversity and setup unity under computation budget. Within each table, the setup is *consistent*, allowing comparison of SL and CL; across tables, we intentionally evaluate if the observation is *generalizable* across backbones and datasets. For example, Tables 1 and 2 are the *same* corruptions but varying backbones; Table B.3 extends the *same* observation from small-scale in Table 3 to larger-scale. The $\Delta$ metric could be unreliable when the original uncorrupted accuracy differs too much across methods. We overcame it by: (1) controlling the original accuracy to be relatively close, (2) testing multiple datasets, backbones, and corruption settings to draw consistent conclusions from more data points.

# 5 ANALYSIS

## 5.1 CL's HIGHER DOWNSTREAM ROBUSTNESS IS RELATED TO A MORE UNIFORM AND STABLE FEATURE SPACE DURING TRAINING

The robustness discrepancy between CL (e.g., MoCo) and SL is not only reflected in the final trained models, but is in fact also attributed in the training process. To analyze how the feature space evolves during training, we measure the following three metrics: **1. Feature Semantic Fluctuation.** We monitor the classification ability of the feature extractor by the accuracy of a KNN probe. We define feature semantic fluctuation of class $i$ as the total variation of per-class accuracy of class $i$ (as a function of epoch $t$) averaged over all epochs: $\mathcal{TV}_i = \frac{1}{T-1} \sum_{t=0}^{T-2} |\text{Acc}_{t+1}^{(i)} - \text{Acc}_t^{(i)}|$. We further define the mean feature semantic fluctuation as the mean of $\mathcal{TV}_i$ over all classes. Larger semantic fluctuation indicates less stable feature space. **2. Feature Uniformity.** We can measure the uniformity of all the features or class-wise features as the log-mean of Gaussian potentials of the normalized features: $U(f_t, \mathcal{D}) = -\log \mathbb{E}_{x_0, x_1 \sim \mathcal{D}} \left[ e^{-2\|f_t(x_0) - f_t(x_1)\|_2^2} \right]$. Here $f_t$ is the network at epoch $t$, $\mathcal{D}$ is the dataset, and $x_0$ and $x_1$ are images sampled from the dataset. The use of this measure to study contrastive learning is exemplified in (Wang & Isola, 2020). Intuitively, a greater $U$ means more uniformly distributed features on the unit sphere, while a smaller value means more concentrated features. **3. Feature Distance.** We also measure the average feature squared $\ell_2$ distance between two classes. A larger distance could mean more linear separability. Denoting $\mathcal{D}_i$ and $\mathcal{D}_j$ as feature matrices of two classes, the fea-

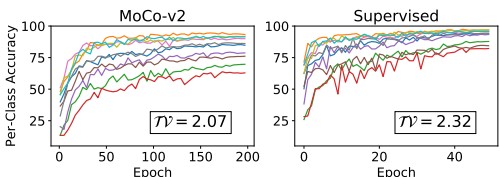

Figure 2: Class-wise test accuracy of MoCo and SL on original CIFAR-10 during training. MoCo has more steady class-wise accuracy curves and smaller mean feature semantic fluctuation ($\mathcal{TV}$) than SL.

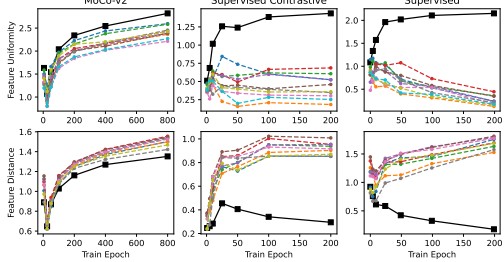

Figure 3: **Above:** Solid black line – uniformity of the overall feature space. Dashed lines – class-wise feature uniformities of the 10 classes. While the overall uniformity of all methods grows, the uniformity of each class of Sup or SupCon is shrinking as training progresses. In the end, the overall uniformity of MoCo is the largest. **Below:** Solid black line – $d(f_t, \mathcal{D}_0, \mathcal{D}_0)$, i.e., the intra-class variance of class 0. Dashed lines – feature distances between $\mathcal{D}_i (i \neq 0)$ and $\mathcal{D}_0$. The intra-class variance behavior of MoCo (increasing) is the opposite to that of Sup or SupCon (decreasing).

Table 7: Uniformity regularization directly influences supervised pre-training's downstream robustness (ResNet-18, CIFAR-10, 200 epochs). Positive uniformity term leads to higher KNN evaluation accuracy on corrupted data with no loss on original accuracy. Subtracting the uniformity term leads to the opposite.

| Pre-train Loss | Metric | Orig | $\gamma 5$ | L4x4 | G4x4 |
|---|---|---|---|---|---|
| Sup | Acc, Unif | 94.18, 1.98 | 72.85, 1.64 | 37.85, 0.98 | 39.70, 0.90 |
| Sup+0.01Unif | Acc, Unif | 94.21, 2.69 | 74.47, 2.03 | 42.22, 1.11 | 44.34, 1.30 |
| Sup−0.01Unif | Acc, Unif | 94.56, 1.12 | 71.50, 0.77 | 36.15, 0.41 | 37.88, 0.46 |

ture distance is calculated as: $d(f_t, \mathcal{D}_i, \mathcal{D}_j) = \mathbb{E}_{x_0 \sim \mathcal{D}_i, x_1 \sim \mathcal{D}_j} \left[ \|f_t(x_0) - f_t(x_1)\|_2^2 \right]$. Note that if $\mathcal{D}_i = \mathcal{D}_j$, it actually measures the intra-class variance of class $i$.

We train ResNet-18 (He et al., 2016) on the original CIFAR-10 (Krizhevsky et al., 2009) train split and measure the above metrics on the test split. Figure 3 shows the dynamics of feature uniformity and distances of MoCo-v2 (He et al., 2020; Chen et al., 2020d), supervised contrastive (SupCon) (Khosla et al., 2020), and supervised learning. We are interested in SupCon, because it bridges CL and SL by leveraging a similar contrastive loss. As illustrated, the overall feature uniformity of MoCo-v2 (Chen et al., 2020d) is greater than 2.5 and approaching 3, while the overall uniformity of SupCon and supervised methods range from 1.25 to 2.2. This means that features from CL methods are more uniformly distributed on the unit sphere. By looking at the class-wise feature uniformity and distance, we notice that SL tends to compress (and maybe over-compress) the features of each class. Figure 2 shows that the accuracy of a KNN probe during supervised learning also fluctuates more dramatically. We can interpret it as that the classes are competing with each other, and SL cannot improve the performance on all classes at the same time like CL methods.

We hypothesize that uniformity is the key to CL's higher downstream robustness, because, intuitively, a more uniform feature space may capture richer characteristics of images and gives the pre-trained model a higher chance to extract useful representation from downstream images, corrupted or not. We test this hypothesis by checking whether SL can benefit from an extra uniformity-promoting loss term. Table 7 briefly demonstrates that adding (or subtracting) the uniformity regularization produces a more (or less) uniform test feature space. This experiment suggests that we could improve SL by leveraging loss functions from CL and potentially get the best of both worlds.

## 5.2 CL's LOWER PRE-TRAINING ROBUSTNESS MAY RELATE TO HIGHER DEPENDENCY ON IMAGE SPATIAL COHERENCE

The diverging robustness behaviors of CL to pre-training corruptions can stem from its higher dependence on image spatial structure. While little previous work examines corruptions during pre-training and its reliance on spatial information, we hypothesize that a high-frequency corruption signals applied globally to the data will harm the long-scale coherence. Such effect is intuitively straight-forward since the authentic spatial information will be eroded and the weighted importance will decrease with the introduction of corruptive information. Table B.8 demonstrates how shuffling interferes with data augmentation in the CIFAR-10 pre-training case. While standard shuffling produces largest avg $\Delta = 22.0\%$, reversing the order of corruption and augmentation greatly ameliorates the $\Delta$ of CL and produces comparable robustness to SL with $\Delta = 6.0\%$. As we perform augmentation such as random resized crop after shuffling, we may select crop windows that capture pieces from different shuffled patches and do not reside in natural image statistics. To better view the destructive effect, Figure C.4 shows the attention maps of global shuffling comparing to the original. Contrastive pre-training with shuffled data leads to less dense and inaccurate attentions, essentially fails to learn good representations, which verifies CL's worse robustness. This also hints that contrastive learning is not really general – on certain types of images it fails. How to design general CL algorithms that work on all kinds of images remains an interesting question.

## 6 CONCLUSION

Our paper systematically studies the distributional robustness of CL and SL through a diverse set of multi-level data corruptions. We discover interesting robustness behaviors of CL to different corruptions. Our analysis of the feature space gives insight that uniformity might be the key to higher downstream robustness, while analyzing augmentation process and attention maps disclose the high dependence of contrastive learning on spatial information. Our results favor the current use of CL or a combination of CL and SL in visual representation learning, and calls for more research into understanding the behavior and the learning mechanism of CL.

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

# A  ADDITIONAL IMPLEMENTATION DETAILS

Table A.1 below lists the experiment configurations for each pre-training robustness table of the main paper. We train our own ResNets (He et al., 2016) on CIFAR-10 (Krizhevsky et al., 2009) and ImageNet variants (Deng et al., 2009). ImageNet-LT/UF are the long-tail and uniformly-subsampled versions. ImageNet-100 is a 100 class subset of full ImageNet-1K. We mainly list Sup and MoCo-v2 (Chen et al., 2020d) hyper-parameters here. The other CL methods follow their recommended hyper-parameter values in the Solo-Learn package (da Costa et al., 2022).

Table A.1: Implementation details for the pre-training results in the main paper.

| Config | Tables 3,5 | Table 6 | Table B.3 |
|---|---|---|---|
| Pre-train dataset | CIFAR-10 | ImageNet-LT/UF | ImageNet-100 |
| # of categories | 10 | 1000 | 100 |
| Train image size | 32 | 224 | 224 |
| Train data size | 50K | 115K | 130K |
| Network | ResNet-18 | ResNet-50 | ResNet-18 |
| Backbone out dim | 512 | 2048 | 512 |
| Sup epochs | 50 | 200 | 50 |
| Sup lr | 0.1 cos | 0.015 cos | 0.015 cos |
| Sup batch size | 512 | 128 | 128 |
| MoCo epochs | 200 | 200 | 200 |
| MoCo lr | 0.06 cos | 0.015 cos | 0.03 cos |
| MoCo batch size | 512 | 128 | 256 |
| MoCo dim | 128 | 128 | 128 |
| MoCo temp. | 0.1 | 0.2 | 0.2 |
| MoCo momentum | 0.99 | 0.999 | 0.999 |
| MoCo queue size | 4096 | 65536 | 65536 |
| Evaluation | Linear | Linear | Linear |
| Augmentation | crop+flip+ color(.4,p=.8) +gray(p=.2) | crop+flip+ color(.4,p=.8)+gray(p=.2)+ gauss(.1,.2,p=.5) | |

# B  ADDITIONAL RESULTS

## B.1  PRE-TRAINING ROBUSTNESS TEST WITH TRANSFORMER BACKBONE

In the main paper, we compare pre-training robustness with a CNN backbone in Table 3, and show Vision Transformer (ViT) downstream robustness test results in Table 2. Here, we supplement ViT *pre-training* robustness test results. Specifically, we leverage MoCo-v3 (Chen et al., 2021b), the ViT version of MoCo, and Supervised ViT. The results are in Table B.1. We find that the MoCo-v3 degradation is larger with patch shuffling, but smaller with gamma distortion. Interestingly, the impact of patch shuffling is much smaller than a CNN (despite the Orig performance gap between ViT and CNN). We suspect that this is due to the unique patching and attention network structure of ViT. Essentially, if we do not take into consideration the data augmentation, with the right patch size, the shuffling within a small patch does not affect the learning of ViT much, and the global ordering of patches also does not matter much, because of learned positional embeddings and global attention.

Table B.1: Pre-training robustness with ViT on CIFAR10: MoCo-v3 vs. Sup. For the ViT architecture, since the input size (32x32) is smaller than that of a standard ViT, we use a customized small ViT (image size=32, patch size=4, dim=512, depth=6, heads=8, mlp dim=512, dropout=0.1, emb dropout=0.1).

| Method | Orig | G4x4 | G16x16 | L4x4 | L16x16 | Avg $\Delta$ | $\gamma = 0.1$ |
|---|---|---|---|---|---|---|---|
| Sup ViT 50ep | 67.92 | 59.01 | 47.97 | 57.76 | 67.95 | - | 52.96 |
| $\Delta$ | - | 13.12% | 29.37% | 14.96% | -0.04% | 14.35% | 22.03% |
| MoCo-v3 200ep | 62.78 | 53.36 | 41.58 | 53.52 | 61.77 | - | 51.41 |
| $\Delta$ | - | 15.0% | 33.77% | 14.75% | 1.61% | 16.28% | 18.11% |

### B.2 Pre-training robustness test with longer epochs

In Table 3 of the main paper, we mostly report results of short pre-training schedules: Sup 30 epochs and CL 200 epochs, in order to make the baseline results comparable. We report CIFAR-10 longer training epochs in Table B.2. Training longer does not change our observation that MoCo appears less robust to patch- and pixel-level corruptions than SL during pre-training on this dataset.

Table B.2: Pre-training robustness: Sup 50ep vs. MoCo-v2 400ep, ResNet-18, CIFAR-10.

| Method | Orig | G4x4 | G8x8 | L4x4 | L8x8 | Avg $\Delta$ | $\gamma = 0.1$ |
|---|---|---|---|---|---|---|---|
| Sup 50ep | 92.23 | 81.14 | 71.33 | 71.72 | 81.95 | - | 89.11 |
| $\Delta$ | - | 12.02% | 22.67% | 22.24% | 11.15% | 17.02% | 3.38% |
| MoCo-v2 400ep | 91.43 | 70.99 | 64.25 | 66.56 | 81.51 | - | 83.94 |
| $\Delta$ | - | 22.36% | 29.73% | 27.20% | 10.85% | 22.54% | 8.19% |

### B.3 Pre-training robustness on ImageNet-100

Table B.3: Robustness Test II: *pre-training pixel- and patch-level* corruptions of ImageNet100. We focus our comparison on MoCo-v2 and SL to train on corrupted ImageNet100, which is a 100-class subset of ImageNet and substantially larger than CIFAR. SL still shows higher robustness to our pixel-level and patch-level corruptions, in agreement with Table 3.

| IN-100 | Orig | $\gamma 0.2$ | $\gamma 5$ | G2x2 | G4x4 | G8x8 | L128x128 | L64x64 | L32x32 | Avg $\Delta \downarrow$ |
|---|---|---|---|---|---|---|---|---|---|---|
| Sup | 77.08 | 73.60 (4.5%) | 70.28 (8.8%) | 67.26 (12.7%) | 62.84 (18.5%) | 58.20 (24.5%) | 75.28 (2.3%) | 72.68 (5.7%) | 68.52 (11.1%) | 6.65% |
| MoCo-v2 | 74.38 | 66.80 (10.2%) | 62.74 (15.6%) | 44.90 (39.6%) | 35.84 (51.8%) | 30.94 (58.4%) | 69.34 (6.8%) | 63.68 (14.4%) | 54.24 (27.1%) | 28.0% |

### B.4 Downstream robustness test with KNN: accuracy numbers

Table B.4 shows the detailed accuracy numbers for computing the summary statistics in Table 1 of the main paper.

### B.5 Downstream robustness test with full fine-tuning

Table 1 in the main paper and Table B.4 above are generated with the KNN evaluation protocol. We also experiment with full fine-tuning on the downstream datasets. The results are in Table B.5. Since different pre-trained checkpoints are optimized with different optimizers (SGD for Sup, SimSiam(Chen & He, 2021), MoCo-v2(Chen et al., 2020d), and SimCLR-v2(Chen et al., 2020c); LARS(You et al., 2017) for BYOL(Grill et al., 2020), BarlowTwins(Zbontar et al., 2021), DeepCluster2, and SwAV(Caron et al., 2020)), we use SGD (lr 0.002 cosine) for Sup, SimSiam, MoCo, and SimCLR, and AdamW (lr 0.001 cosine) (Loshchilov & Hutter, 2019) for others during fine-tuning. All models are fine-tuned for 10 epochs. We find this strategy of using different optimizers is able to make the baseline results on original images comparable across methods. We note that fine-tuning drastically improves the accuracy on downstream datasets, while the general observation that CL methods are more robust to downstream corruption than SL still holds, except for BarlowTwins which is slightly worse than SL. Another interesting observation here is that *different CL methods actually yield different robustness behaviors*, although they are all doing some form of contrastive learning and have similar baseline accuracies.

### B.6 Variance of pre-training results

We repeat MoCo-v2 on the original CIFAR-10 200ep three times: The KNN evaluation mean and std is $82.44 \pm 0.18$. Repeating MoCo-v2 on the global 8x8 shuffling corrupted CIFAR-10 gives KNN evaluation mean and std $59.24 \pm 0.40$. The linear evaluation variance is similar. The randomness has a smaller order than the gap between MoCo and Sup results.

### B.7 Pre-train on corrupted CIFAR-10, but test on uncorrupted images

In the main paper, we show the results when both the pre-training and evaluation datasets are corrupted in the same consistent way. In the following Table B.7, we report the accuracy numbers obtained from KNN evaluation on the original uncorrupted images. Since these models are pre-trained on the pixel- or patch-level corrupted dataset, the results reflect the transfer capability of the

Table B.4: Robustness to downstream data corruption with *KNN evaluation*. This table contains the detailed top-1 accuracy numbers constituting Table 1 in the main paper. The shades of yellow in the last column indicate the size of the numbers. Suffix '-b' of an algorithm refers to pre-trained model from another source.

| Pre-train Alg | Dataset | Orig | $\gamma = 0.2$ | $\gamma = 5$ | G-small | G-large | L-small | L-large | Avg $\Delta$ |
|---|---|---|---|---|---|---|---|---|---|
| Sup | cifar10 | 86.4 | 76.4 (11.7%) | 67.6 (21.8%) | 61.2 (29.2%) | 49.6 (42.6%) | 46.9 (45.7%) | 53.6 (38.0%) | 31.5% |
| Sup | cifar100 | 65.1 | 52.4 (19.6%) | 44.1 (32.3%) | 37.4 (42.6%) | 25.7 (60.5%) | 23.4 (64.1%) | 30.8 (52.8%) | 45.3% |
| Sup | stl10 | 96.6 | 92.2 (4.6%) | 82.6 (14.5%) | 80.8 (16.3%) | 40.7 (57.8%) | 43.4 (55.1%) | 60.3 (37.5%) | 31.0% |
| Sup | cars196 | 26.8 | 23.3 (13.2%) | 18.1 (32.5%) | 12.7 (52.7%) | 4.4 (83.6%) | 4.1 (84.9%) | 16.0 (40.4%) | 51.2% |
| Sup | aircraft70 | 40.3 | 37.9 (6.0%) | 38.8 (3.6%) | 25.2 (37.4%) | 8.0 (80.3%) | 10.0 (75.2%) | 25.5 (36.7%) | 39.9% |
| Sup-b | cifar10 | 84.9 | 77.0 (9.3%) | 68.9 (18.9%) | 57.7 (32.0%) | 46.5 (45.2%) | 44.0 (48.2%) | 51.9 (38.9%) | 32.1% |
| Sup-b | cifar100 | 63.2 | 53.3 (15.7%) | 45.4 (28.1%) | 33.0 (47.8%) | 21.4 (66.1%) | 18.9 (70.0%) | 28.0 (55.7%) | 47.2% |
| Sup-b | stl10 | 96.0 | 92.8 (3.3%) | 83.6 (12.9%) | 77.9 (18.9%) | 36.0 (62.5%) | 41.5 (56.7%) | 60.4 (37.0%) | 31.9% |
| Sup-b | cars196 | 28.8 | 26.8 (7.0%) | 22.0 (23.8%) | 12.1 (58.1%) | 1.8 (93.7%) | 2.4 (91.6%) | 15.8 (45.2%) | 53.2% |
| Sup-b | aircraft70 | 46.9 | 47.0 (-0.3%) | 46.0 (1.9%) | 29.8 (36.4%) | 7.7 (83.5%) | 10.8 (76.9%) | 29.8 (36.5%) | 39.2% |
| BYOL | cifar10 | 87.5 | 80.3 (8.3%) | 72.4 (17.3%) | 64.0 (26.8%) | 50.7 (42.1%) | 48.4 (44.7%) | 55.6 (36.5%) | 29.3% |
| BYOL | cifar100 | 67.4 | 58.1 (13.8%) | 49.8 (26.0%) | 39.7 (41.1%) | 26.1 (61.3%) | 24.5 (63.6%) | 32.1 (52.3%) | 43.0% |
| BYOL | stl10 | 94.9 | 92.4 (2.6%) | 85.0 (10.4%) | 78.1 (17.7%) | 41.5 (56.3%) | 43.8 (53.8%) | 63.0 (33.6%) | 29.0% |
| BYOL | cars196 | 22.5 | 22.3 (0.8%) | 18.7 (17.0%) | 12.6 (44.1%) | 2.5 (88.9%) | 2.3 (89.6%) | 18.6 (17.2%) | 42.9% |
| BYOL | aircraft70 | 38.2 | 39.5 (-3.3%) | 38.6 (-1.2%) | 24.0 (37.2%) | 8.0 (79.2%) | 10.5 (72.6%) | 31.1 (18.5%) | 33.8% |
| SimSiam | cifar10 | 83.6 | 77.6 (7.2%) | 68.7 (17.8%) | 61.5 (26.5%) | 50.4 (39.8%) | 48.4 (42.2%) | 55.6 (33.5%) | 27.8% |
| SimSiam | cifar100 | 59.9 | 52.7 (12.1%) | 44.8 (25.1%) | 36.1 (39.7%) | 24.8 (58.6%) | 23.2 (61.2%) | 31.1 (48.0%) | 40.8% |
| SimSiam | stl10 | 92.1 | 89.6 (2.7%) | 81.1 (12.0%) | 74.2 (19.4%) | 39.1 (57.6%) | 43.8 (52.5%) | 62.7 (31.9%) | 29.3% |
| SimSiam | cars196 | 17.1 | 15.9 (6.9%) | 13.9 (18.5%) | 10.3 (39.4%) | 2.0 (88.6%) | 2.5 (85.2%) | 15.3 (10.2%) | 41.5% |
| SimSiam | aircraft70 | 32.8 | 34.1 (-3.7%) | 32.6 (0.8%) | 20.9 (36.3%) | 6.8 (79.3%) | 10.8 (67.2%) | 27.6 (15.9%) | 32.6% |
| MoCo | cifar10 | 81.4 | 74.0 (9.1%) | 66.1 (18.9%) | 60.2 (26.0%) | 49.5 (39.2%) | 47.2 (42.0%) | 54.0 (33.7%) | 28.1% |
| MoCo | cifar100 | 56.6 | 48.2 (14.8%) | 41.9 (26.0%) | 35.3 (37.7%) | 23.5 (58.5%) | 23.1 (59.3%) | 30.0 (46.9%) | 40.5% |
| MoCo | stl10 | 90.1 | 88.1 (2.2%) | 77.5 (13.9%) | 73.5 (18.5%) | 39.9 (55.7%) | 42.8 (52.5%) | 60.0 (33.4%) | 29.4% |
| MoCo | cars196 | 13.1 | 12.8 (2.5%) | 11.3 (14.2%) | 8.8 (32.9%) | 2.3 (82.6%) | 2.6 (80.5%) | 12.1 (8.2%) | 36.8% |
| MoCo | aircraft70 | 25.2 | 26.7 (-6.1%) | 23.4 (7.2%) | 17.3 (31.5%) | 8.1 (67.9%) | 10.0 (60.2%) | 21.2 (15.7%) | 29.4% |
| MoCo-b | cifar10 | 83.6 | 76.1 (9.0%) | 67.1 (19.7%) | 57.2 (31.5%) | 47.2 (43.5%) | 45.6 (45.5%) | 51.0 (38.9%) | 31.3% |
| MoCo-b | cifar100 | 59.5 | 49.9 (16.1%) | 42.4 (28.7%) | 32.9 (44.7%) | 21.4 (64.0%) | 21.1 (64.5%) | 27.9 (53.0%) | 45.2% |
| MoCo-b | stl10 | 95.3 | 92.8 (2.6%) | 85.1 (10.6%) | 76.0 (20.2%) | 38.9 (59.2%) | 40.4 (57.6%) | 60.9 (36.1%) | 31.0% |
| MoCo-b | cars196 | 13.8 | 13.7 (1.3%) | 12.1 (12.8%) | 7.9 (42.7%) | 1.8 (86.9%) | 1.9 (86.3%) | 12.7 (8.4%) | 39.7% |
| MoCo-b | aircraft70 | 26.8 | 28.1 (-4.9%) | 26.4 (1.5%) | 17.2 (35.8%) | 6.9 (74.4%) | 8.6 (67.8%) | 23.3 (13.2%) | 31.3% |
| SimCLR2 | cifar10 | 85.4 | 79.2 (7.3%) | 67.5 (21.0%) | 58.0 (32.1%) | 45.6 (46.6%) | 45.8 (46.4%) | 54.8 (35.8%) | 31.5% |
| SimCLR2 | cifar100 | 63.5 | 55.2 (13.1%) | 44.6 (29.8%) | 33.2 (47.7%) | 21.2 (66.7%) | 22.4 (64.7%) | 31.4 (50.6%) | 45.4% |
| SimCLR2 | stl10 | 91.9 | 89.3 (2.9%) | 81.7 (11.2%) | 69.8 (24.1%) | 38.4 (58.2%) | 40.8 (55.6%) | 61.5 (33.1%) | 30.8% |
| SimCLR2 | cars196 | 17.7 | 16.9 (4.6%) | 15.6 (11.9%) | 9.8 (44.5%) | 1.6 (91.2%) | 2.4 (86.7%) | 14.3 (19.1%) | 43.0% |
| SimCLR2 | aircraft70 | 31.2 | 32.0 (-2.4%) | 32.0 (-2.3%) | 20.1 (35.7%) | 7.1 (77.4%) | 10.2 (67.2%) | 26.7 (14.6%) | 31.7% |
| BarlowTwins | cifar10 | 83.8 | 77.8 (7.1%) | 70.0 (16.4%) | 62.0 (26.0%) | 51.6 (38.5%) | 50.0 (40.3%) | 56.9 (32.1%) | 26.7% |
| BarlowTwins | cifar100 | 63.7 | 56.0 (12.1%) | 48.1 (24.5%) | 38.8 (39.2%) | 26.5 (58.5%) | 26.6 (58.2%) | 34.2 (46.3%) | 39.8% |
| BarlowTwins | stl10 | 94.5 | 91.6 (3.0%) | 83.7 (11.4%) | 74.6 (21.1%) | 40.0 (57.7%) | 44.8 (52.6%) | 63.7 (32.6%) | 29.7% |
| BarlowTwins | cars196 | 23.4 | 23.6 (-1.1%) | 20.7 (11.4%) | 11.8 (49.6%) | 2.7 (88.4%) | 2.4 (89.8%) | 18.7 (19.9%) | 43.0% |
| BarlowTwins | aircraft70 | 39.2 | 43.1 (-9.7%) | 40.4 (-2.9%) | 22.5 (42.7%) | 7.7 (80.3%) | 9.4 (76.1%) | 31.4 (19.9%) | 34.4% |
| DeepCluster | cifar10 | 87.2 | 80.5 (7.7%) | 70.6 (19.0%) | 64.3 (26.2%) | 52.5 (39.7%) | 50.3 (42.3%) | 57.3 (34.3%) | 28.2% |
| DeepCluster | cifar100 | 65.0 | 56.2 (13.6%) | 47.3 (27.1%) | 39.6 (39.1%) | 27.7 (57.4%) | 25.6 (60.6%) | 33.5 (48.5%) | 41.1% |
| DeepCluster | stl10 | 94.8 | 92.4 (2.6%) | 84.6 (10.8%) | 79.2 (16.5%) | 41.9 (55.8%) | 45.0 (52.6%) | 64.0 (32.5%) | 28.5% |
| DeepCluster | cars196 | 22.7 | 20.9 (7.8%) | 19.3 (15.0%) | 13.8 (39.3%) | 2.9 (87.2%) | 3.8 (83.2%) | 16.7 (26.4%) | 43.2% |
| DeepCluster | aircraft70 | 40.3 | 39.2 (2.8%) | 37.6 (6.7%) | 24.4 (39.5%) | 7.3 (81.8%) | 11.2 (72.2%) | 27.9 (30.7%) | 38.9% |
| SwAV | cifar10 | 83.5 | 76.8 (8.1%) | 68.8 (17.7%) | 63.3 (24.1%) | 52.9 (36.6%) | 48.6 (41.8%) | 55.3 (33.8%) | 27.0% |
| SwAV | cifar100 | 60.1 | 52.5 (12.7%) | 44.3 (26.3%) | 38.5 (35.9%) | 27.1 (55.0%) | 23.8 (60.4%) | 31.0 (48.4%) | 39.8% |
| SwAV | stl10 | 94.4 | 91.8 (2.8%) | 84.0 (11.1%) | 80.3 (14.9%) | 43.0 (54.5%) | 44.5 (52.8%) | 62.7 (33.6%) | 28.3% |
| SwAV | cars196 | 17.2 | 16.2 (5.8%) | 14.6 (15.0%) | 12.0 (30.4%) | 3.0 (82.8%) | 3.0 (82.5%) | 12.6 (26.9%) | 40.6% |
| SwAV | aircraft70 | 31.5 | 29.7 (5.6%) | 30.5 (3.0%) | 23.9 (24.2%) | 8.3 (73.7%) | 10.3 (67.2%) | 22.1 (29.7%) | 33.9% |
| SwAV-b | cifar10 | 84.7 | 78.0 (7.9%) | 70.1 (17.2%) | 63.4 (25.1%) | 52.6 (37.8%) | 51.0 (39.8%) | 56.7 (33.0%) | 26.8% |
| SwAV-b | cifar100 | 62.7 | 54.4 (13.2%) | 46.5 (25.8%) | 39.7 (36.6%) | 28.0 (55.3%) | 26.5 (57.8%) | 33.2 (47.1%) | 39.3% |
| SwAV-b | stl10 | 94.3 | 91.6 (2.9%) | 83.7 (11.3%) | 78.5 (16.8%) | 44.6 (52.8%) | 45.0 (52.4%) | 60.9 (35.4%) | 28.6% |
| SwAV-b | cars196 | 19.3 | 18.0 (6.8%) | 16.4 (15.2%) | 12.4 (36.1%) | 3.1 (84.2%) | 3.5 (82.0%) | 14.7 (24.2%) | 41.4% |
| SwAV-b | aircraft70 | 33.5 | 32.8 (2.1%) | 30.5 (9.0%) | 21.7 (35.2%) | 8.0 (76.0%) | 11.0 (67.2%) | 23.9 (28.7%) | 36.3% |

Table B.5: Robustness to downstream data corruption with *fine-tuning*. We fine-tune the full network and linear classification layer for 10 epochs. Overall, CL methods are more robust than Sup under this setting except for BarlowTwins. Suffix '-b' of an algorithm refers to pre-trained model from another source. We include algorithm average Δ for each algorithm across all datasets for a clearer comparison.

| Pre-train Alg / Alg Avg | Dataset | Orig | $\gamma = 0.2$ | $\gamma = 5$ | G-small | G-large | L-small | L-large | Avg Δ |
|---|---|---|---|---|---|---|---|---|---|
| Sup | cifar10 | 96.7 | 96.8 (-0.2%) | 94.0 (2.8%) | 88.2 (8.8%) | 77.5 (19.8%) | 72.0 (25.5%) | 86.0 (11.0%) | 11.3% |
| Sup | cifar100 | 83.8 | 83.7 (0.1%) | 77.4 (7.6%) | 68.8 (17.9%) | 53.5 (36.2%) | 46.0 (45.1%) | 65.2 (22.2%) | 21.5% |
| Sup | stl10 | 97.7 | 97.2 (0.6%) | 92.5 (5.4%) | 92.1 (5.7%) | 55.5 (43.2%) | 56.5 (42.2%) | 89.1 (8.8%) | 17.6% |
| Sup | cars196 | 75.1 | 73.2 (2.6%) | 58.4 (22.3%) | 40.1 (46.5%) | 4.1 (94.6%) | 5.0 (93.4%) | 56.5 (24.7%) | 47.3% |
| Sup | aircraft70 | 81.5 | 80.4 (1.3%) | 78.8 (3.3%) | 66.5 (18.4%) | 11.5 (85.9%) | 17.6 (78.4%) | 72.8 (10.6%) | 33.0% |
| Alg Avg Δ | 26.1% | | | | | | | | |
| BYOL | cifar10 | 96.5 | 96.3 (0.2%) | 93.9 (2.7%) | 88.8 (7.9%) | 80.2 (16.9%) | 75.2 (22.0%) | 87.4 (9.4%) | 9.8% |
| BYOL | cifar100 | 83.2 | 82.2 (1.2%) | 76.7 (7.8%) | 68.3 (17.9%) | 54.2 (34.8%) | 46.5 (44.1%) | 64.4 (22.6%) | 21.4% |
| BYOL | stl10 | 96.2 | 95.8 (0.5%) | 91.8 (4.7%) | 91.3 (5.2%) | 57.0 (40.8%) | 56.2 (41.6%) | 88.2 (8.4%) | 16.9% |
| BYOL | cars196 | 80.4 | 77.2 (4.0%) | 62.1 (22.8%) | 49.0 (39.1%) | 2.8 (96.6%) | 3.9 (95.2%) | 65.1 (19.0%) | 46.1% |
| BYOL | aircraft70 | 87.7 | 86.5 (1.4%) | 84.1 (4.2%) | 76.3 (13.0%) | 13.9 (84.1%) | 20.2 (76.9%) | 80.0 (8.8%) | 31.4% |
| Alg Avg Δ | 25.1% | | | | | | | | |
| SimSiam | cifar10 | 95.0 | 95.1 (-0.1%) | 92.1 (3.1%) | 87.5 (7.9%) | 79.8 (16.1%) | 75.4 (20.7%) | 86.7 (8.8%) | 9.4% |
| SimSiam | cifar100 | 81.0 | 80.5 (0.6%) | 74.1 (8.5%) | 68.5 (15.5%) | 56.4 (30.4%) | 51.3 (36.6%) | 67.1 (17.1%) | 18.1% |
| SimSiam | stl10 | 94.0 | 93.4 (0.6%) | 88.1 (6.3%) | 87.1 (7.3%) | 64.4 (31.5%) | 59.5 (36.7%) | 85.8 (8.7%) | 15.2% |
| SimSiam | cars196 | 85.7 | 85.2 (0.6%) | 75.7 (11.6%) | 64.4 (24.9%) | 4.2 (95.1%) | 5.9 (93.1%) | 79.3 (7.5%) | 38.8% |
| SimSiam | aircraft70 | 89.7 | 89.0 (0.8%) | 86.9 (3.2%) | 82.3 (8.3%) | 23.5 (73.8%) | 28.9 (67.7%) | 86.4 (3.7%) | 26.3% |
| Alg Avg Δ | 21.6% | | | | | | | | |
| MoCo-b | cifar10 | 96.8 | 96.7 (0.2%) | 94.5 (2.4%) | 89.6 (7.5%) | 81.5 (15.9%) | 77.4 (20.1%) | 89.4 (7.7%) | 9.0% |
| MoCo-b | cifar100 | 84.8 | 84.1 (0.9%) | 78.5 (7.5%) | 72.2 (14.8%) | 59.0 (30.5%) | 53.9 (36.4%) | 70.8 (16.5%) | 17.8% |
| MoCo-b | stl10 | 96.3 | 96.3 (0.0%) | 91.8 (4.6%) | 91.6 (4.9%) | 64.3 (33.2%) | 61.0 (36.7%) | 90.3 (6.2%) | 14.3% |
| MoCo-b | cars196 | 85.7 | 84.6 (1.3%) | 75.7 (11.7%) | 62.8 (26.7%) | 3.6 (95.8%) | 5.0 (94.2%) | 78.5 (8.3%) | 39.7% |
| MoCo-b | aircraft70 | 90.3 | 89.3 (1.1%) | 88.0 (2.5%) | 82.5 (8.6%) | 22.6 (75.0%) | 27.2 (69.9%) | 86.9 (3.8%) | 26.8% |
| Alg Avg Δ | 21.5% | | | | | | | | |
| SimCLR2 | cifar10 | 96.3 | 95.8 (0.5%) | 93.3 (3.1%) | 87.1 (9.6%) | 76.8 (20.3%) | 72.2 (25.0%) | 86.2 (10.5%) | 11.5% |
| SimCLR2 | cifar100 | 84.8 | 84.2 (0.7%) | 78.6 (7.3%) | 69.5 (18.1%) | 56.7 (33.2%) | 51.4 (39.4%) | 67.3 (20.7%) | 19.9% |
| SimCLR2 | stl10 | 95.5 | 95.2 (0.3%) | 89.7 (6.0%) | 86.5 (9.4%) | 54.6 (42.8%) | 55.8 (41.6%) | 88.0 (7.8%) | 18.0% |
| SimCLR2 | cars196 | 77.9 | 75.3 (3.4%) | 64.9 (16.8%) | 47.0 (39.6%) | 3.0 (96.2%) | 4.5 (94.3%) | 68.1 (12.6%) | 43.8% |
| SimCLR2 | aircraft70 | 84.8 | 83.8 (1.1%) | 82.9 (2.2%) | 72.5 (14.5%) | 20.1 (76.3%) | 23.8 (72.0%) | 79.4 (6.3%) | 28.7% |
| Alg Avg Δ | 24.4% | | | | | | | | |
| BarlowTwins | cifar10 | 96.8 | 96.7 (0.1%) | 94.4 (2.5%) | 87.9 (9.2%) | 76.4 (21.0%) | 70.1 (27.6%) | 84.9 (12.3%) | 12.1% |
| BarlowTwins | cifar100 | 83.9 | 83.6 (0.4%) | 76.9 (8.4%) | 64.2 (23.5%) | 46.1 (45.1%) | 39.0 (53.5%) | 56.4 (32.8%) | 27.2% |
| BarlowTwins | stl10 | 97.3 | 96.8 (0.6%) | 92.2 (5.2%) | 91.2 (6.3%) | 52.9 (45.7%) | 52.0 (46.6%) | 87.1 (10.5%) | 19.1% |
| BarlowTwins | cars196 | 73.5 | 69.0 (6.3%) | 53.2 (27.7%) | 38.0 (48.3%) | 2.7 (96.4%) | 3.4 (95.3%) | 57.1 (22.4%) | 49.4% |
| BarlowTwins | aircraft70 | 81.1 | 77.9 (4.0%) | 76.5 (5.7%) | 63.8 (21.3%) | 11.3 (86.0%) | 15.5 (80.9%) | 67.7 (16.5%) | 35.7% |
| Alg Avg Δ | 28.7% | | | | | | | | |
| DeepCluster2-b | cifar10 | 96.5 | 96.5 (0.0%) | 94.6 (2.0%) | 89.9 (6.9%) | 80.8 (16.3%) | 75.7 (21.6%) | 87.7 (9.2%) | 9.3% |
| DeepCluster2-b | cifar100 | 84.7 | 83.6 (1.3%) | 78.3 (7.5%) | 71.6 (15.4%) | 57.6 (32.0%) | 49.2 (41.9%) | 66.8 (21.1%) | 19.9% |
| DeepCluster2-b | stl10 | 96.8 | 96.3 (0.4%) | 93.7 (3.2%) | 93.1 (3.8%) | 62.3 (35.6%) | 57.6 (40.4%) | 88.9 (8.1%) | 15.3% |
| DeepCluster2-b | cars196 | 81.6 | 79.4 (2.6%) | 68.6 (16.0%) | 56.3 (31.0%) | 3.4 (95.8%) | 4.9 (94.0%) | 66.5 (18.5%) | 43.0% |
| DeepCluster2-b | aircraft70 | 87.9 | 87.2 (0.8%) | 85.4 (2.9%) | 77.8 (11.5%) | 15.4 (82.5%) | 20.0 (77.2%) | 79.1 (10.1%) | 30.8% |
| Alg Avg Δ | 23.7% | | | | | | | | |
| SwAV-b | cifar10 | 96.3 | 96.4 (-0.2%) | 94.0 (2.3%) | 89.8 (6.7%) | 81.6 (15.2%) | 75.9 (21.2%) | 87.8 (8.7%) | 9.0% |
| SwAV-b | cifar100 | 83.7 | 83.1 (0.7%) | 77.4 (7.5%) | 70.8 (15.4%) | 58.1 (30.5%) | 49.7 (40.6%) | 66.3 (20.8%) | 19.3% |
| SwAV-b | stl10 | 96.3 | 96.6 (-0.3%) | 92.8 (3.7%) | 92.7 (3.8%) | 63.2 (34.3%) | 58.9 (38.9%) | 88.6 (8.0%) | 14.7% |
| SwAV-b | cars196 | 82.2 | 80.1 (2.5%) | 70.5 (14.2%) | 60.4 (26.5%) | 3.8 (95.4%) | 5.4 (93.4%) | 67.7 (17.6%) | 41.6% |
| SwAV-b | aircraft70 | 89.2 | 88.2 (1.2%) | 87.2 (2.3%) | 80.0 (10.4%) | 18.2 (79.6%) | 22.9 (74.3%) | 81.3 (8.9%) | 29.5% |
| Alg Avg Δ | 22.8% | | | | | | | | |

pre-trained representation from corrupted data to original data. We find that the trend is similar to evaluating on corrupted data that Sup appears more robust.

Table B.7: Uncorrupted evaluation results of robustness to pre-training pixel-level gamma distortion and patch-level corruption (global and local shuffling) with CIFAR-10 and ResNet-18.

| Method | Orig | $\gamma = 0.2$ | G4x4 | G8x8 | L4x4 | L8x8 | Avg $\Delta$ |
|---|---|---|---|---|---|---|---|
| Sup | 92.23 | 82.72 | 63.03 | 36.94 | 61.56 | 62.51 | - |
| $\Delta$ | - | 10.31% | 31.66% | 59.95% | 33.25% | 32.22% | 33.48% |
| MoCo-v2 KNN | 82.55 | 72.01 | 46.66 | 32.93 | 48.91 | 53.78 | - |
| $\Delta$ | - | 15.40% | 43.48% | 60.11% | 40.75% | 34.85% | 38.39% |

## B.8 REVERSING CORRUPTION-AUGMENTATION ORDER

While our study is based on learning corrupted data, we can switch the corruption-augmentation order for deeper analysis. Table B.8 shows the shuffling case on CIFAR-10 during pre-training. The standard shuffling corruption on MoCo leads to the largest avg $\Delta = 22.0\%$, while switching the order brings comparable robustness to supervised learning in terms of avg $\Delta$. This provides insight on the different dependence between CL and SL on spatial information corrupted by shuffling.

Table B.8: Additional pre-training corruption with Sup no-augmentation and Sup/MoCo augmentation-then-corrupt variants. SL is able to learn without data augmentation. Contrary to the corrupt-aug version in previous sections, MoCo and Sup share roughly a similar level of robustness with the aug-corrupt order.

| Pre-training | Orig. | G4x4 | G8x8 | L4x4 | L8x8 | Avg $\Delta$ |
|---|---|---|---|---|---|---|
| Sup no-aug | 87.66 | 77.37 (11.7%) | 71.86 (18.0%) | 73.30 (16.4%) | 82.34 (6.0%) | 13.0% |
| Sup aug-corrupt | 92.23 | 85.92 (6.8%) | 80.58 (12.6%) | 83.61 (9.4%) | 89.96 (2.5%) | 7.8% |
| MoCo corrupt-aug | 82.55 | 65.43 (17.1%) | 59.49 (27.9%) | 59.62 (27.8%) | 70.14 (15.0%) | 22.0% |
| MoCo aug-corrupt | 82.55 | 77.63 (6.0%) | 73.48 (11.0%) | 78.12 (5.4%) | 81.25 (1.6%) | 6.0% |

## B.9 MEASURING THE TASK DIFFICULTY BY H-DIVERGENCE

To demonstrate that different corruptions at different levels of strength have corresponding different levels of difficulty, we have quantified the H-divergence as shown in Figure B.1.

We follow Lemma 2 from the H-divergence paper (Ben-David et al., 2010) and implement the objective using PyTorch. Since the objective is straightforward and a convolutional network converges very fast to zero loss (and H divergence 2), we adopt multilayer perceptron (MLP) with sigmoid function to observe the progress and differences between data corruptions. We select two different strengths for each of our proposed data corruption and observe stronger corruptions are indeed proportionally farther from the original dataset with higher H divergence. The selected data corruptions from ImageNet-C (Hendrycks & Dietterich, 2019) cannot be distinguished well by our MLP. We can resolve it with a deeper convolutional network and longer training, but it suffices to say that ImageNet-C provides mild corruptions, which also corresponds to the smaller performance drop shown in Table 4. We do not evaluate dataset level corruptions since class imbalance already changes class distributions and GAN-synthesized dataset is trained to minimize divergence. To empirically verify the different dynamics of feature space, we have adopted a few metrics to evaluate the feature distance and uniformity, and quantify them for CL and SL models at each epoch to discuss the progress throughout pre-training as shown in Figure 2 and Figure 3. As we have mentioned in Sec. 3.2, we also select a few natural data corruptions from ImageNet-C, which together with other corruptions contribute to our final conclusions.

## C ADDITIONAL VISUALIZATION

### C.1 VISUALIZING GRAD-CAM ATTENTION MAPS

Figure C.1 visualizes the Grad-CAM (Selvaraju et al., 2017) attention maps of ResNet-18 models pre-trained and linearly fine-tuned on either uncorrupted or 4x4 global patch shuffled images. We discover some difference in terms of the *equivariant* property: Sup models are largely equivariant to 4x4 global patch shuffling – the attention is focused on the object parts even after patch shuffling, whereas the MoCo model pre-trained on 4x4 global shuffled images are not – it is rather focused

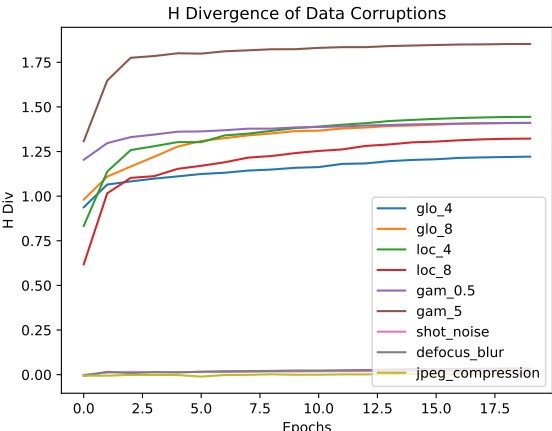

Figure B.1: H-divergence between the original dataset and the corrupted dataset as measured by training a simple network to distinguish them.

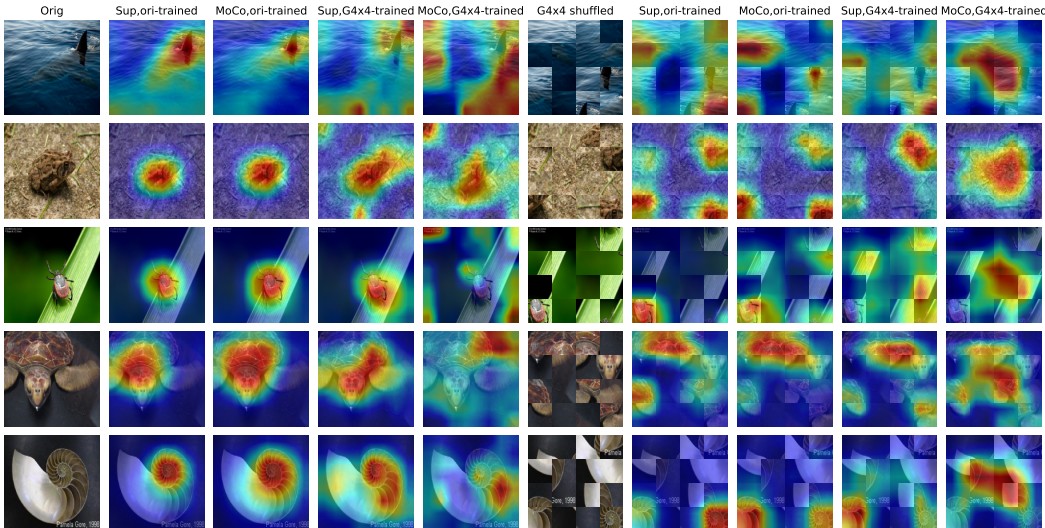

Figure C.1: Randomly chosen images from ImageNet-100. We consider 4x4 global patch shuffling and visualize the Grad-CAM attention maps of 4 models: Sup trained on original images, MoCo trained on original images, Sup trained on shuffled images, and MoCo trained on shuffled images. The attention map of the MoCo model on shuffled images is less equivariant to the patch shuffling.

on distracting parts. The quality of attention maps correlates with the top-1 validation accuracy, where Sup on 4x4 achieves 65% and MoCo achieves 35%. Intuitively, a model can be more robust to the global patch shuffling if it possesses such an equivariant property. This shows the robustness of SL from another aspect, because it can robustly learn the same feature even under the shuffling disturbance.

To further understand qualitatively how different corruption strategies impact the model's ability to learn semantic concepts, we draw the CAMs of models trained under different corruption settings on the corrupted versions of two ImageNet validation images in Figures C.2 and C.3. Global shuffling and defocus blur especially hinder the ability of MoCo to learn meaningful semantics.

## C.2  VISUALIZING CORRUPTED IMAGES

Please check Figure C.4 for more visual examples of the pixel-level gamma distortion and patch-level shuffling corruptions we used.

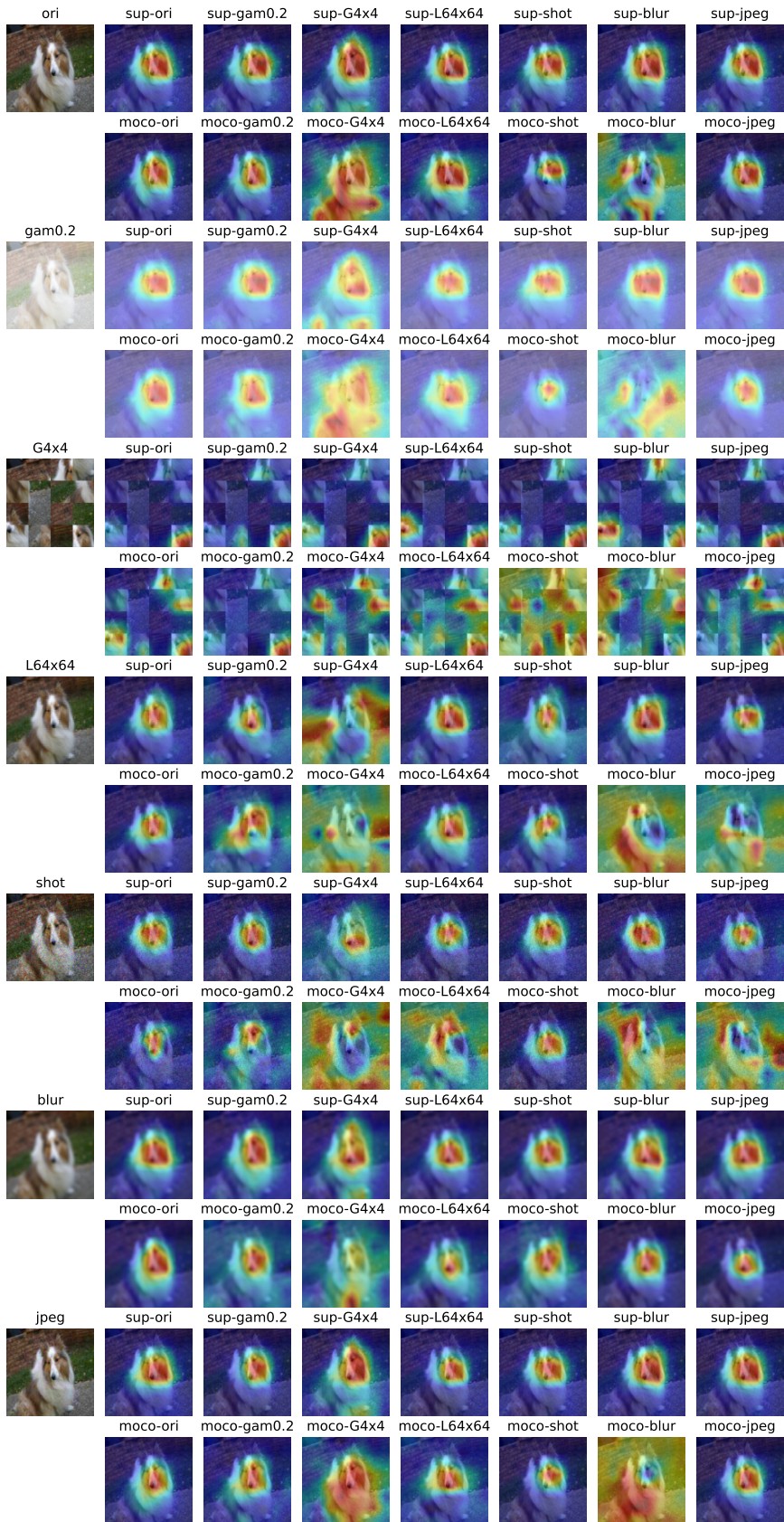

Figure C.2: GradCAM on corrupted versions of a dog image of sup/MoCo models trained under 7 corruptions.

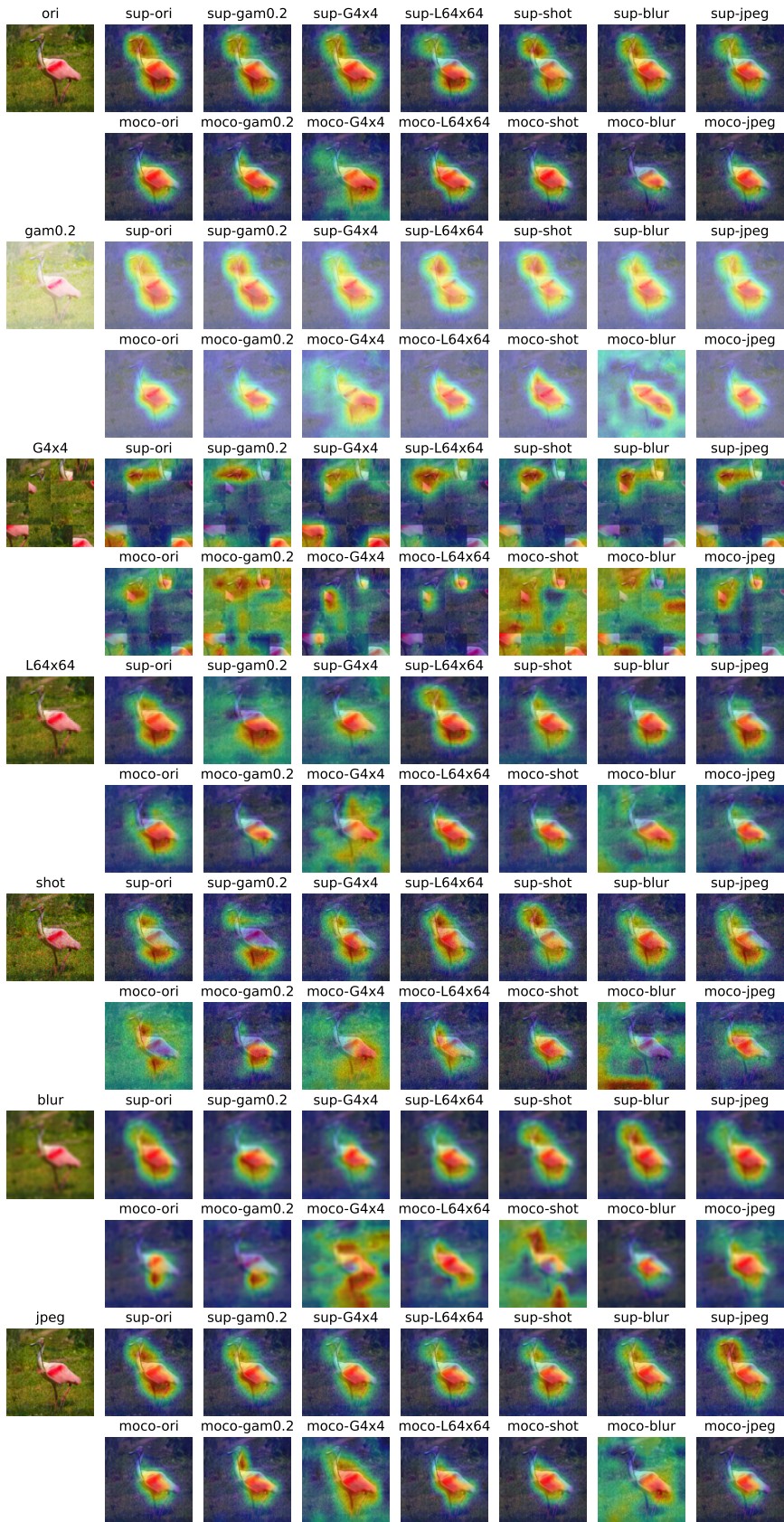

Figure C.3: GradCAM on corrupted versions of a bird image of sup/MoCo models trained under 7 corruptions.

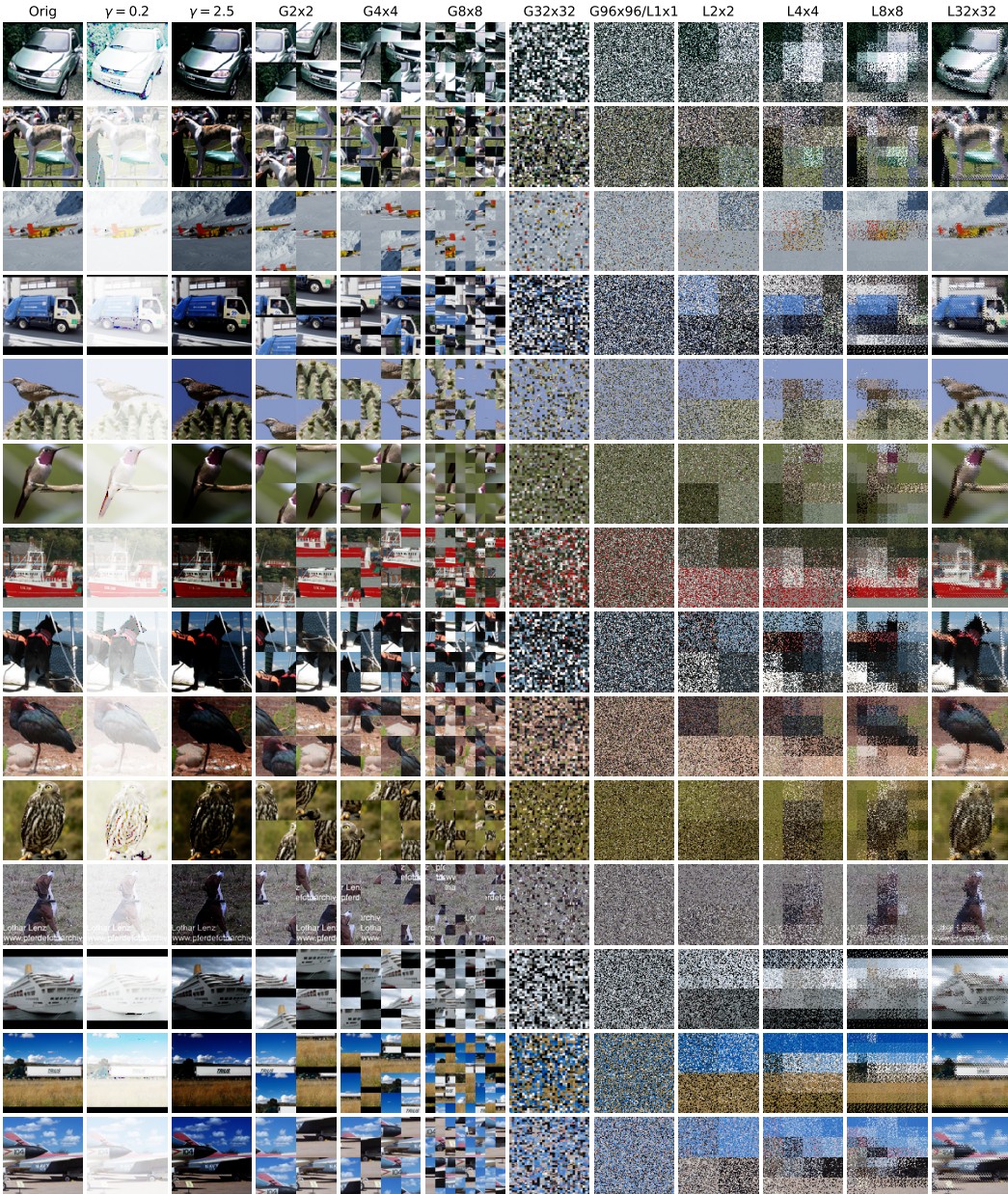

Figure C.4: Randomly chosen examples from the STL-10 dataset. The original images have resolution 96x96. We show the resulting images of gamma distortion ($\gamma = 0.2, 2.5$), global shuffling (G2x2, weaker – G96x96, stronger), and local shuffling (L1x1, stronger – L32x32, weaker). G1x1 and L96x96 revert to the original, while G96x96 and L1x1 are the most random ones (and have similar effect). Gamma distortion reduces information in pixel intensity. Global shuffling destroys global but preserves local structure, while local shuffling is the opposite.

