# OpenReview forum: "Is Self-Supervised Contrastive Learning More Robust Than Supervised Learning?"
_ICLR.cc/2023/Conference — Submitted to ICLR 2023_

### Official Review · Reviewer_CG2S · 2022-10-14

**Confidence:** 3
**Correctness:** 3
**Technical Novelty And Significance:** 2
**Empirical Novelty And Significance:** 2
**Recommendation:** 3

**Clarity, Quality, Novelty And Reproducibility:**

Clarity: As discussed above, the write-up could improve in terms of defining upfront some terms that often overloaded in our community (e.g., robustness).

Quality: The empirical evaluation seems to have been well executed. Nonetheless, concerns remain in terms of the underlying motivations for such an assessment and the limitations in the conclusions themselves. In particular, it's hard to tell how the conclusions change depending on the "amount of shift", and there's no analysis indicating why the observed behaviours manifest as they do; i.e., what is it that causes the different behaviours?  Authors do hypothesize the nature of the training objective is the reason, since contrastive methods tend less to collapse to a subspace of the feature space, but that hypothesis should be verified empirically since there's recent work arguing contrastive losses yield embeddings of incomplete rank [3].

Novelty: The current version of the manuscript is limited in terms of novelty since it's already known that both considered learning schemes suffer from covariate shift, and that self-supervision seems to suffer less. I would say that extensions of the experiments showing why that is would be novel and informative to the overall community.

Reproducibility: The experiments setup are clearly described and should be easy to reproduce from the details in the text.

[3] Jing L, Vincent P, LeCun Y, Tian Y. Understanding dimensional collapse in contrastive self-supervised learning. arXiv preprint arXiv:2110.09348. 2021 Oct 18.

**Strength And Weaknesses:**

Pros:

+The empirical assessment covers a number of datasets and seems to consistently justify the posed conclusions.

+A diverse set of data corruption processes is considered.

Cons:

-Unclear motivation: It's unclear to me after reading through the manuscript what is the motivation for measuring this notion of robustness the authors posed. Most of the corruption approaches considered are unnatural, so we should not expect conclusions to hold similarly once we move to naturally occurring variations in the data. I would recommend including an explicit motivation statement in the text. To be explicit, the question I could not get an answer for from the text is: what do we get from determining whether supervision or self-supervision is less affected by unnatural corruptions?

-Missing related work: Robustness against distribution shifts has been largely studied in recent years and novel work should contextualize itself with respect to prior literature. It seems to me, for instance, that authors specifically consider the covariate shift cases where data marginals change (via perturbations), but class conditional distributions are unchanged. It would be then important to state that in the text, and further mention that conclusions hold for that setting only. There's also been prior work showing self-supervision to observe greater robustness to covariate shift than supervision, such as [1].

-Limited empirical assessment: The empirical evaluation has some limitations that should be mentioned. Conclusions do not control for task difficulty, and it's unclear whether varying levels of distortion impose varying levels of difficulty. I would suggest measuring some notion of distance between every considered dataset and corrupted versions, so one can get a sense of different behaviours against different levels of difficulty. The H-divergence [2] would be a natural candidate metric for such an analysis, but something else could be used as well. Moreover, the evaluation focuses on determining performance gaps, but lacks in explaining those. Authors hypothesize the main reason for differences across supervision and self-supervision are due to the effective dimension of the features, but that should be empirically verified. Finally, I would also suggest considering naturally occurring data perturbations.

-Clarity: I would say the write up could benefit of stating early on what is meant by terms that are usually overloaded in the literature. The main issue in my opinion is the usage of 'robustness'. There are several notions of robustness currently in use, so a definition should appear as early as possible. The same goes for somewhat generic terms such as stability and behaviour.

[1] Albuquerque I, Naik N, Li J, Keskar N, Socher R. Improving out-of-distribution generalization via multi-task self-supervised pretraining. arXiv preprint arXiv:2003.13525. 2020 Mar 30.

[2] Ben-David S, Blitzer J, Crammer K, Kulesza A, Pereira F, Vaughan JW. A theory of learning from different domains. Machine learning. 2010 May;79(1):151-75.

**Summary Of The Paper:**

This paper presents an empirical assessment of variations in behaviour of different representation learning paradigms when the underlying training sample is corrupted with noise. Authors focused on image classification tasks, and verified gap in performances post corruptions. They further evaluated against which kinds of corruptions each of supervised or self-supervised contrastive schemes observes the greater drop in prediction accuracy, and that analysis is carried out both when only classification heads are trained downstream, as well as when  training the complete embedding encoder. Authors further discussed metrics that can be measured from the representations across layers and that seem to correlate with performance gap.

**Summary Of The Review:**

While the paper carefully carried out a somewhat large scale evaluation, I would say the current version still lacks some extra analysis in order for it to be conclusive (detailed above).

---

> ### Author Response · Authors · 2022-11-21
> **Response to Reviewer CG2S [Part 1]**
>
> We sincerely thank you for your valuable suggestions. Below we clarify the confusions and provide further explanations to your questions.
>
> #### 1. [General clarification on novelty]
>
> Before we address the specific concerns, we would like to first clarify the major concern of novelty, in the following aspects:
>
> * New settings
>
> We want to emphasize that, in addition to the downstream data corruption, our study also investigates **pre-training data corruption**.
>
> The two settings characterize two complementary aspects of distributional robustness. The downstream corruption studies how a pre-trained representation space copes with distribution-shifted data. The pre-training corruption studies how the learning algorithm performs with distribution-shifted training data.
>
> Our settings are novel, especially the pre-training corruption setting. As noted by you and Reviewer J9U1, most distributional robustness papers only focus on the *downstream* setting with *uncorrupted* data. A systematic comparison of SL and CL to a wide range of *pre-training* corruptions is missing in the literature.
>
> * Comprehensive corruption types
>
> Our data corruptions contain pixel-level (manipulating pixel values), patch-level (destroying global or local structural information), and dataset-level (changing dataset distribution), for us to comprehensively evaluate the impact of corrupted data in the downstream, and how such corruptions interfere with feature learning. Such comprehensiveness does not exist in prior works, thus making our conclusions more general and well-supported.
>
> Importantly, we consider **both natural and unnatural** corruptions. Most of the corruption types are natural: shot-noise, blur, compression, gamma distortion, as well as class imbalance and synthetic data, and they can all happen in real life. The patch shuffling corruption is unnatural, but we specifically choose this corruption to study whether breaking down the image spatial structural information affects feature learning.
>
> * New findings
>
> 1) While previous works reveal better robustness of CL than SL against out-of-domain and class distribution shifts, such an observation is limited to a few settings. We significantly enrich this observation to a wider set of corruption types and models, covering a hierarchy of corruption levels and 7 different contrastive learning algorithms.
>
> 2) Under-explored in the community and to the best of our knowledge, we are the first to observe that CL is **not more robust** than SL during pre-training for many cases, despite it producing more robust pre-trained models. In particular, we discover that CL really relies much more crucially than SL on the image spatial structure.
>
> * Explanation of the findings
>
> As mentioned in the submission, we claim our paper as an empirical study. While a thorough theoretical analysis is beyond the scope of this paper, we do hypothesize different reasons for the pre-training and the downstream robustness differences, and we have empirically validated these hypotheses.
>
> In the downstream, CL is more robust – We hypothesize the main reason is uniformity. We have measured the uniformity empirically in Sec. 5.1, and verified that CL indeed has higher uniformity than SL. We have also conducted an experiment showing increasing the feature uniformity during SL can improve SL’s downstream robustness.
>
> In the pre-training stage, CL is less robust than SL to pixel and patch-level corruptions. We hypothesize this may be due to the higher dependency of CL’s data augmentation strategy on the image structure in Sec. 5.2. To verify this, we show that the hypothetical setting of switching the order of augmentation and corruption significantly reduces the accuracy drop of CL.
>
> To sum up, we believe that our new findings based on comprehensive evaluations are of sufficient interest to the field. We hope they can inspire future work to study similar problems (e.g., behavioral difference of learning algorithms), provide comprehensive explanations to the discovered phenomenons, and ultimately work towards fully understanding CL.

---

> ### Author Response · Authors · 2022-11-21
> **Response to Reviewer CG2S [Part 2]**
>
> #### 2. [Motivation for unnatural corruptions]
>
> We apologize for the confusion. We clarify our motivation below.
>
> First of all, we believe that it is inaccurate to say that “most of the corruption approaches are unnatural.” We **actually investigated many natural corruptions** as shown in Figure 1. The purpose is to know how the natural corruptions affect the algorithm’s learning (pre-training robustness) or the pretrained model’s ability to transfer (downstream robustness) in realistic settings.
>
> Specifically, the pixel-level corruptions are natural: shot-noise, defocus blur, and gamma distortion mimic what could happen to a real camera – due to the photon sensor noise and imperfect lens parameters; jpeg compression happens everyday in internet transmission. The dataset-level corruptions are also natural: everyday object categories are long-tail distributed; tons of synthetic data are being posted on the internet (e.g., stable diffusion).
>
> Measuring robustness to such natural corruptions are practically relevant. We found that CL is generally more robust than SL to those corruptions in the downstream stage, however less robust to naturally-occuring pixel corruptions in the pre-training stage.
>
> In addition, we investigated the unnatural patch-shuffling corruptions. The purpose is to understand how CL and SL algorithms and models behave when we destroy the natural image spatial structure. The observation is generally consistent with that from natural corruptions, but the accuracy drop of CL on globally patch-shuffled pre-training data is much larger. To answer your question, what we get from measuring the robustness to unnatural corruption is **discovering that CL really relies much more crucially than SL on the image spatial structure**. We hypothesize this is because the coherent spatial structure is what makes the common cropping data augmentation appropriate for CL (Sec. 5.2).
>
> #### 3. [Related work]
>
> Thank you for your pointer. Yes, our study majorly concerns distributional shifts of input images (at pixel, patch, and dataset level via multi-level data corruptions), and we only corrupt on class distribution when evaluating class imbalance (unbalanced long-tail distribution vs. balanced distribution). The most relevant investigation is ImageNet-C (Hendrycks & Dietterich, 2019), which only contains a set of natural corruptions overlaying on the original data such as weather, noise points, and blurring. We have cited and sufficiently discussed this work in sections of related work and methodology. Similarly, compared with other related work which we have discussed, the investigation in our paper is much more comprehensive, thus making our conclusions more general and well-supported. The insight from [1] in fact agrees with our observations on downstream tasks where CL learns more robust representations, **but it does not involve any dataset corruption nor the pre-training stage**. We have now included and discussed this work in the related work of the revision.
>
> #### 4. [Quantify task difficulty]
>
> Thank you for the great suggestion! Following your suggestion, to demonstrate that different corruptions at different levels of strength have corresponding different levels of difficulty, we have quantified the H-divergence [2] on p.18 (Section B.9 and Figure B.1) of the revision. In particular, we find that the shot-noise, defocus blur, and jpeg corrupted data are close to the original data distribution, hence easier. Gamma distortion (with extreme parameters) and patch shuffling deviate a lot from original distribution, hence harder. It aligns well with our empirical performance – We do see that the performance drop on latter ones is larger. With patch shuffling, more global pieces (h-div glo_8 > glo_4) and less local pieces (loc_4 < loc_8) correspond to higher degree of disorder (xref Fig. C.4), which is intuitive.

---

> ### Author Response · Authors · 2022-11-21
> **Response to Reviewer CG2S [Part 3]**
>
> #### 5. [Explain the performance gaps]
>
> As mentioned in the general response above, we hypothesize **different reasons for the pre-training and the downstream robustness differences**. In the downstream, CL is more robust – We hypothesize the main reason is uniformity, assuming by “effective dimension of the features” the reviewer refers to the uniformity metric since we have not used “effective dimension” in our paper. We have measured the uniformity empirically in Sec. 5.1, and verified that CL indeed has higher uniformity than SL. We have also conducted an experiment showing increasing the feature uniformity during SL can improve SL’s downstream robustness.
>
> In the pre-training stage, CL is less robust than SL to pixel and patch-level corruptions. We hypothesize this may be due to the higher dependency CL’s data augmentation strategy on image structure in Sec. 5.2. To verify this, we show that the hypothetical setting of switching the order of augmentation and corruption significantly reduces the accuracy drop of CL.
>
> #### 6. [Definition of terms]
>
> Thank you for your suggestion! We have modified the first paragraph of Section 3 “Method” to clarify our definitions of distributional robustness and behavior of an algorithm, before we explain our method and experiment in detail.
>
> [1] Albuquerque I, Naik N, Li J, Keskar N, Socher R. Improving out-of-distribution generalization via multi-task self-supervised pretraining. arXiv preprint arXiv:2003.13525. 2020 Mar 30.
>
> [2] Ben-David S, Blitzer J, Crammer K, Kulesza A, Pereira F, Vaughan JW. A theory of learning from different domains. Machine learning. 2010 May;79(1):151-75.

---

> > ### Comment · Reviewer_CG2S · 2022-12-02
> > **Response to authors.**
> >
> > Thank you for the clarifications and additions to the manuscript.
> >
> > After going through responses and comments by other reviewers, I still have concerns and will keep my original score for now. Main concerns are as follows:
> >
> > - I still don't see a clear motivation as to why we would measure this notion of robustness at the pre-training stage. While the authors clarified that some of the corruptions naturally occur, we should then expect that those would be already present in a realistic dataset. Apart from class imbalance, it's unclear to me how this analysis can be informative.
> >
> > - The downstream robustness behaviour is inline with past findings, which limits novelty/utility of the reported evidence as well.
> >
> > - As pointed out by Rev. fxue, hyperparameters should be adjusted once the underlying data source changes otherwise the analysis accounts for suboptimal models in the corrupted data.

---

> > > ### Author Response · Authors · 2022-12-10
> > > **Response to follow-up questions [Part 1]**
> > >
> > > We thank you for your genuine suggestions, and would like to further clarify your concerns.
> > >
> > > **1. [Motivation for measuring robustness at the pre-training stage]**
> > >
> > > We would like to further clarify our motivations for measuring robustness at the pre-training stage from the following two aspects.
> > >
> > > (1) A **key motivation** is to understand CL algorithms: **what data information or image structure is critical for the high performance of CL algorithms**. By default, most of the existing CL algorithms are pre-trained on **natural, curated, class-balanced data** like ImageNet. Given that CL algorithms have no access to class semantic labels but they still learn good feature representations, a fundamental question arises: why is this the case? Our insight then is that the superior performance of CL algorithms might be due to the fact that they implicitly exploit certain image structures or information inherent in the existing datasets as inductive bias. What if we break the clean data assumption? This leads to our empirical investigation towards answering the question.
> > >
> > > To this end, we need to purposefully corrupt the pre-training data to deviate from the conventional assumption like natural images and class-balanced distribution and even produce unnatural images that do not occur in the existing curated datasets. Specifically, we systematically design the types of corruptions that destroy different aspects of image information: pixel-level – pixel intensity statistics, patch-level – image spatial structure, dataset-level – overall instance distribution. Our study shows that for several kinds of data corruption (e.g., pixel-level corrupted, patch-shuffled images), CL works quite poorly. This further **indicates that CL algorithms potentially highly rely on, for example, strong natural image spatial structure as inductive bias for achieving good performance, compared with supervised learning; once such spatial structure does not hold due to patch-level shuffling, their performance drops significantly**.
> > >
> > > (2) In addition, we respectfully disagree with the reviewer that “we should expect that the corruptions would be already present in a realistic dataset.” We would like to argue that, due to the careful data curation process when constructing a dataset, most of the existing datasets are still not at the level of faithfully reflecting the real world. For example, on ImageNet the objects are typically centered in the images. Therefore, we cannot expect that the corruption should be already present in these datasets.
> > >
> > > On the other hand, in practice, there is often little control over pre-training data. This situation is akin to adversarial attacks, which also introduce natural/unnatural perturbations to the real data. Newly emerging models scrape images of mixed quality from the Internet (e.g., CLIP [1]), and more unsecured data will be adopted for pre-training, which is potentially corrupted by various kinds of corruption attacks. Therefore, our study raises valid concerns over such emerging issues in pre-training; and indeed, our study reveals the significantly decreased performance of CL algorithms with corrupted data – This corresponds to our **second motivation of being aware of the potential failure of existing CL algorithms and when**: CL algorithms are not as general as we might have thought; instead, they are actually customized to ImageNet-like, natural, object-centric images.
> > >
> > > [1] Alec Radford, et al. Learning transferable visual models from natural language supervision. In ICML, 2021.

---

> > > ### Author Response · Authors · 2022-12-10
> > > **Response to follow-up questions [Part 2]**
> > >
> > > **2. [Downstream robustness]**
> > >
> > > Although the observation of downstream robustness is in line with past findings, we enrich it and make significant novel contributions with respect to (a) focusing on different types of downstream robustness from prior works and investigating comprehensive and diverse corruption settings, (b) providing more understanding through uniformity, and (c) contrasting it against pre-training robustness. More specifically,
> > >
> > > * [Robustness to downstream corruptions] Our work enriches our understanding of CL with a significantly broader scope of downstream robustness. Our downstream robustness is different from prior works. Previous works focus on other aspects of downstream robustness, such as adversarial and out-of-domain datasets. They work with **uncorrupted data** as opposed to **corrupted data** in our case. It was not obvious that the past findings could carry over without doing the experiments. Our experiments with diverse corruptions have filled this gap. We find CL is also more robust than SL to corruptions of downstream datasets.
> > >
> > > * [Understanding through uniformity] Our work also provides a novel explanation about why CL is more robust than SL in downstream corruption. We hypothesize the more uniform feature space of CL directly **relates to** the more downstream *corruption* robustness, and design the experiment in Sec. 5.1 Table 7 to verify our hypothesis.
> > >
> > > * [Contrasting with pre-training] Moreover, another novel finding is the contrasting observation of CL **between pre-training and downstream robustness**, which was not previously known. CL is actually less robust to pre-training pixel/patch corruptions than SL.
> > >
> > > **3. [Hyper-parameters]**
> > >
> > > We thank the reviewer for the suggestion. As mentioned in the previous response, the change of hyper-parameter settings will not affect our current outcomes. We have provided additional validation in the following table, which tunes the hyper-parameters of MoCo-v2 on each corrupted CIFAR10 setting as in Table 3. For a fair comparison, we also provide the result of tuning the hyper-parameters of supervised learning on each corrupted setting. We searched various sets of hyper-parameters over a wide range, including CL-specific terms such as temperature. **The improvements with tuning hyper-parameters are marginal, and our conclusions remain the same as in the submission**. We will include these results in the revision.
> > >
> > > | Model | gam 0.2 | glo 4 | glo 8 | loc 4 | loc 8 | avg delta |
> > > |:---:|:---:|:---:|:---:|:---:|:---:|:---:|
> > > | sup (89.53) submission | 87.36 (2.4%) | 76.06 (15.0%) | 65.88 (26.4%) | 65.94 (26.3%) | 77.49 (13.4%) | 16.7% |
> > > | sup-tuned | 87.46 (2.3%) | 76.90 (14.1%) | 66.24 (26.0%) | 66.43 (25.8%) | 77.93 (13.0%) | 16.2% |
> > > | MoCo-v2 (88.73) submission | 85.84 (3.3%) | 67.18 (24.3%) | 60.51 (31.8%) | 63.35 (28.6%) | 76.90 (13.3%) | 20.3% |
> > > | MoCo-v2-tuned | 86.17 (2.9%) | 67.92 (23.5%) | 62.70 (29.3%) | 63.81 (28.1%) | 77.07 (13.1%) | 19.4% |
> > >
> > > We look forward to your feedback, and we will further revise our manuscript with our additional clarifications.

---

### Official Review · Reviewer_J9U1 · 2022-10-24

**Confidence:** 5
**Correctness:** 4
**Technical Novelty And Significance:** 2
**Empirical Novelty And Significance:** 2
**Recommendation:** 5

**Clarity, Quality, Novelty And Reproducibility:**

This paper has clarified clearly. The sufficient experiment with analysis and visualization can fit for the discovered phenomenon.

However, this paper is not much novel for the analysis and discussion between contrastive learning and supervised learning. Details can be seen on above weakness.


**Strength And Weaknesses:**

Strength:

-This paper investigate an interesting problem, that is, how does changes in data distribution affect the robust behaviors of contrastive and supervised learning?

-The design of data corruption types ranging from micro-level(pixel-level, patch-level) to macro-level(dataset-level) can better evaluate pre-training algorithms’ behavior under various distortions.

Weaknesses:

-This paper has mentioned many aspects of robustness such as: robustness to long-tail or noisy training data, Robustness of the model output, Robustness to input corruptions at test-time. However, in practice this paper only measure accuracy degradation on certain corruption, which seems trivial to some extent. It is better to say, the main phenomenon is CL benefits from some data augmentation more than SL in some cases.

-Contrastive learning methods shows more robustness compared with supervised learning seems to be a global known behavior in transfer learning field, as discussed in [A][B]. Demonstrating this experimental phenomenon is not novel.

-The analysis of contrastive learning from the perspective of uniformity and stable feature space have also been discussed by a widely known paper [C].

-The third contribution that “a simple way to improve the downstream robustness of supervised learning”, which adding an additional loss as contrastive learning to supervised learning is also not novel. For example, the third section of [A] also provide a MoCo-like loss to get a better supervised pretraining.

[A] Nanxuan Zhao, Zhirong Wu, Rynson WH Lau, and Stephen Lin. What makes instance discrimination good for transfer learning? In ICLR, 2021.

[B] Linus Ericsson, Henry Gouk, and Timothy M Hospedales. How well do self-supervised models transfer? In CVPR, 2021.

[C] Tongzhou Wang and Phillip Isola. Understanding contrastive representation learning through alignment and uniformity on the hypersphere. In ICML, 2020.


**Summary Of The Paper:**

This paper design and conduct a series of robustness tests to quantify the behavioral differences between contrastive learning and supervised learning to downstream and pre-training data distribution changes, and conclude that contrastive learning is more robust than supervised learning under downstream corruptions. This paper also discover that diverging robustness behaviors between CL and SL, and even among different CL algorithms. After that, this paper give analyses and explanations, and provide a simple way to improve the downstream robustness of supervised learning.

**Summary Of The Review:**

This paper has presented some meaningful intuitive experimental results. However, the main motivation that “Contrastive learning methods shows more robustness compared with supervised learning” and the main conclusion “CL’s robustness is related to a more uniform and stable feature space” and improvement of supervised learning by adding a loss seems less novel. From another perspective, some parts of this article are more like a patchwork of some existing theoretical verifications, and lacking their own verifications.

---

> ### Author Response · Authors · 2022-11-21
> **Response to Reviewer J9U1 [Part 1]**
>
> We sincerely thank you for your valuable suggestions. Below we clarify the confusions and provide further explanations to your questions.
>
> #### 1. [General clarification on novelty]
>
> Before we address the specific concerns, we would like to first clarify the major concern of novelty, in the following aspects:
>
> * New settings
>
> We want to emphasize that, in addition to the downstream data corruption, our study also investigates **pre-training data corruption**.
>
> The two settings characterize two complementary aspects of distributional robustness. The downstream corruption studies how a pre-trained representation space copes with distribution-shifted data. The pre-training corruption studies how the learning algorithm performs with distribution-shifted training data.
>
> Our settings are novel, especially the pre-training corruption setting. As noted by you and Reviewer CG2S, most distributional robustness papers only focus on the *downstream* setting with *uncorrupted* data. A systematic comparison of SL and CL to a wide range of *pre-training* corruptions is missing in the literature.
>
> * Comprehensive corruption types
>
> Our data corruptions contain pixel-level (manipulating pixel values), patch-level (destroying global or local structural information), and dataset-level (changing dataset distribution), for us to comprehensively evaluate the impact of corrupted data in the downstream, and how such corruptions interfere with feature learning. Such comprehensiveness does not exist in prior works, thus making our conclusions more general and well-supported.
>
> Importantly, we consider **both natural and unnatural** corruptions. Most of the corruption types are natural: shot-noise, blur, compression, gamma distortion, as well as class imbalance and synthetic data, and they can all happen in real life. The patch shuffling corruption is unnatural, but we specifically choose this corruption to study whether breaking down the image spatial structural information affects feature learning.
>
> * New findings
>
> 1) While previous works reveal better robustness of CL than SL against out-of-domain and class distribution shifts, such an observation is limited to a few settings. We significantly enrich this observation to a wider set of corruption types and models, covering a hierarchy of corruption levels and 7 different contrastive learning algorithms.
>
> 2) Under-explored in the community and to the best of our knowledge, we are the first to observe that CL is **not more robust** than SL during pre-training for many cases, despite it producing more robust pre-trained models. In particular, we discover that CL really relies much more crucially than SL on the image spatial structure.
>
> * Explanation of the findings
>
> As mentioned in the submission, we claim our paper as an empirical study. While a thorough theoretical analysis is beyond the scope of this paper, we do hypothesize different reasons for the pre-training and the downstream robustness differences, and we have empirically validated these hypotheses.
>
> In the downstream, CL is more robust – We hypothesize the main reason is uniformity. We have measured the uniformity empirically in Sec. 5.1, and verified that CL indeed has higher uniformity than SL. We have also conducted an experiment showing increasing the feature uniformity during SL can improve SL’s downstream robustness.
>
> In the pre-training stage, CL is less robust than SL to pixel and patch-level corruptions. We hypothesize this may be due to the higher dependency of CL’s data augmentation strategy on the image structure in Sec. 5.2. To verify this, we show that the hypothetical setting of switching the order of augmentation and corruption significantly reduces the accuracy drop of CL.
>
> To sum up, we believe that our new findings based on comprehensive evaluations are of sufficient interest to the field. We hope they can inspire future work to study similar problems (e.g., behavioral difference of learning algorithms), provide comprehensive explanations to the discovered phenomenons, and ultimately work towards fully understanding CL.

---

> ### Author Response · Authors · 2022-11-21
> **Response to Reviewer J9U1 [Part 2]**
>
> #### 2. [Reason to mention many robustness concepts]
>
> First, we want to clarify that the reason to mention the "many aspects of robustness" is to emphasize that the bigger picture of robustness behavior has been under-studied in the pre-training field. We list these different aspects in the introduction to raise awareness to such a broad problem space. Our paper then focuses on one critical aspect, which is the algorithm's robustness to pre-training or downstream distribution shifts caused by corruptions. Our investigation on this particular type of robustness is important, because it uncovers previously **unknown** observations (such as the contrasting difference that CL turns out to be less robust than SL to pre-training corruptions, despite being more robust to downstream corruptions), and leads us one step closer to understanding CL better.
>
> [Accuracy degradation seems trivial]
>
> We disagree with the reviewer that our observation is trivial. Our main observation, as the title suggests, is that **the degrees of accuracy degradation between CL and SL are different** and depend on whether the corruption happens in pre-training or downstream. This is not trivial at all.
>
> We find CL is generally more robust than SL to downstream corruptions, **across CL algorithms, diverse corruption types and datasets**. Although existing work has suggested similar observations for out-of-domain data, **it has never been tested on a broad spectrum of corrupted data** such as ours. We hypothesize the more uniform feature space of CL directly relates to the more downstream corruption robustness, and design the experiment in Sec. 5.1 Table 7 to verify it.
>
> Second, perhaps more interestingly, **the observation is the opposite under pre-training pixel and patch-level corruptions**. This time CL is much less robust than SL. This previously unknown phenomenon suggests there is a fundamental difference between the mechanisms of CL and SL, although they may show comparable ImageNet top-1 accuracy.
>
> [It is better to say, the main phenomenon is CL benefits from some data augmentation more than SL in some cases.]
>
> Regarding the reviewer’s statement on "the main phenomenon is CL benefits from some data augmentation more than SL in some cases," we agree this is another way to put it. But (1) this statement is a bit narrow: it is just one part of our findings, and our work has revealed a much higher degree of complexity inside the robustness of SL/CL to pre-training and downstream (as in the general response above), which cannot be simply summarized as a simple conclusion "the main phenomenon is CL benefits from some data augmentation more than SL in some cases." (2) This statement is too conceptual and not precise: Our investigation is much more rigorous with comprehensive quantitative evaluations. Sec. 5.2 reflects the reviewer's point, but in a quantitative, precise manner, that CL relies crucially on the augmentation pipeline carefully-catered to uncorrupted, natural and curated images, and corruption such as global patch shuffling easily destroys such assumption along with CL feature learning process – again, we believe this is our novel finding.
>
> #### 3. [Novelty of experiment compared with [A][B]]
>
> We think there might be some confusion. Our experiment settings are new and are very different from [A] and [B]. [A] and [B] discover that CL models transfer better than SL to **uncorrupted** clean downstream data, and measure the accuracy (not robustness); by contrast, we study whether CL models transfer better than SL to **corrupted** downstream data, and measure the **accuracy drop** compared with transferring to uncorrupted data. We define less accuracy drop as more robustness.
>
> These are related but different settings. They are related in the sense that both compare CL and SL and try to understand CL better. We actually have already cited both [A][B] in the related work and [A] in the introduction of the original submission as representative work of the recent trend of understanding CL. However, they are different because ours concern robustness but theirs don’t. Neither [A] nor [B] has mentioned the word “robustness”.
>
> Our experimental phenomenon is therefore on a different aspect. We discover the accuracy drop of CL models induced by downstream corruption is smaller than SL, hence CL models have higher robustness. In addition to the downstream corruption setting, we also have the pre-training corruption setting, which was not explored in [A] or [B] at all.

---

> ### Author Response · Authors · 2022-11-21
> **Response to Reviewer J9U1 [Part 3]**
>
> #### 4. [Uniformity known in [C]]
>
> Yes, we did not invent uniformity, but drew the definition of uniformity from [C] as a powerful analysis tool and a possible explanation to CL’s higher downstream robustness. On the other hand, our paper is different from [C] in the following important aspects:
>
> [C] shows that the uniformity of the entire feature space of the **final checkpoint** of CL is higher than SL on uncorrupted data. We not only show this, but also show (1) how the uniformity changes **during training** in Figure 3, and (2) the **per-class uniformity trend**: increases in CL, decreases in SL, as in Figure 3.
>
> Finally, the stability comparison of feature space shown in Figure 2 has not been discussed in [C]. We find the semantic classes are more stable throughout CL pre-training than SL, indicated by a less fluctuated classification accuracy curve. This is a different concept from the uniformity discussed in [C].
>
>
> #### 5. [Adding additional loss to SL]
>
> Our third contribution is an intuitive verification to the hypothesis that uniformity matters for downstream robustness. We can improve SL’s distributional robustness to downstream corruptions by adding a feature uniformity term. This differs from [A] since Section 3 of [A] (1) contains both the alignment **and** the uniformity term, as it basically filters out false negatives with labels in the MoCo loss, (2) concerns transferring to uncorrupted downstream data only.
>
> We intend to shed light on future work and applications of our study, but our paper’s main emphasis is on the new observations of different robustness behaviors of CL and SL in train and test time, rather than proposing a new method to improve SL.
>
>
>
> [A] Nanxuan Zhao, Zhirong Wu, Rynson WH Lau, and Stephen Lin. What makes instance discrimination good for transfer learning? In ICLR, 2021.
>
> [B] Linus Ericsson, Henry Gouk, and Timothy M Hospedales. How well do self-supervised models transfer? In CVPR, 2021.
>
> [C] Tongzhou Wang and Phillip Isola. Understanding contrastive representation learning through alignment and uniformity on the hypersphere. In ICML, 2020.

---

### Official Review · Reviewer_XV1R · 2022-10-24

**Confidence:** 3
**Clarity, Quality, Novelty And Reproducibility:** The paper is clear and provides novel…
**Correctness:** 3
**Technical Novelty And Significance:** 2
**Empirical Novelty And Significance:** 2
**Recommendation:** 5

**Strength And Weaknesses:**

Pros:

1. The paper is well-written. I find it easy to follow and understand.

2. The paper does extensive experiments on multiple levels and two types of data corruption.

3. The author provides a nice interpretation of why CL is more robust to downstream corruption by checking the feature uniformity. And leverage this fact to improve the SL's robustness to downstream corruption. This result is novel and interesting.

Cons:

1. Unjustified metric: the paper proposes to use relative accuracy drop (RAP) as the robust metric. I am not convinced why the absolute accuracy drop (AAP) is not chosen here.

2. Artificial Corruption: The author studies patch-level corruption. As the author admits in the paper "Note that patch shuffling is not commonly used in the standard augmentation pipeline". If patch-level corruption is not real and is usually not encountered in real practice, why should we care about the result of patch-level corruption? The author could argue that the study is done for the purpose of understanding CL's behavior. However, I do not see deeper and more intrinsic findings than CL is not robust to patch-level corruption.


**Summary Of The Paper:**

The paper discusses contrastive learning's robustness to data corruption compared with that of supervised learning. Two types of corruption, pertaining to corruption and downstream corruption are considered. A systematic study on multiple datasets is provided.

**Summary Of The Review:**

Overall, I find the paper borderline. On one hand, it provides new findings of CL's robustness to data corruption. However, on the other hand, the paper includes many related studies without well motivating them. For example,  as I mentioned above, why should we study patch shuffling corruption? Why should we use related accuracy drop as the metric?

---

> ### Author Response · Authors · 2022-11-21
> **Response to Reviewer XV1R [Part 1]**
>
> We sincerely thank you for your valuable suggestions and your acknowledgement. Below we clarify the confusion over our metrics and motivations for unnatural corruptions below.
>
> #### 1. [General clarification on novelty]
>
> Before we address the specific concerns, we would like to first clarify our novelty, in the following aspects:
>
> * New settings
>
> We want to emphasize that, in addition to the downstream data corruption, our study also investigates **pre-training data corruption**.
>
> The two settings characterize two complementary aspects of distributional robustness. The downstream corruption studies how a pre-trained representation space copes with distribution-shifted data. The pre-training corruption studies how the learning algorithm performs with distribution-shifted training data.
>
> Our settings are novel, especially the pre-training corruption setting. As noted by Reviewer J9U1 and Reviewer CG2S, most distributional robustness papers only focus on the *downstream* setting with *uncorrupted* data. A systematic comparison of SL and CL to a wide range of *pre-training* corruptions is missing in the literature.
>
> * Comprehensive corruption types
>
> Our data corruptions contain pixel-level (manipulating pixel values), patch-level (destroying global or local structural information), and dataset-level (changing dataset distribution), for us to comprehensively evaluate the impact of corrupted data in the downstream, and how such corruptions interfere with feature learning. Such comprehensiveness does not exist in prior works, thus making our conclusions more general and well-supported.
>
> Importantly, we consider **both natural and unnatural** corruptions. Most of the corruption types are natural: shot-noise, blur, compression, gamma distortion, as well as class imbalance and synthetic data, and they can all happen in real life. The patch shuffling corruption is unnatural, but we specifically choose this corruption to study whether breaking down the image spatial structural information affects feature learning.
>
> * New findings
>
> 1) While previous works reveal better robustness of CL than SL against out-of-domain and class distribution shifts, such an observation is limited to a few settings. We significantly enrich this observation to a wider set of corruption types and models, covering a hierarchy of corruption levels and 7 different contrastive learning algorithms.
>
> 2) Under-explored in the community and to the best of our knowledge, we are the first to observe that CL is **not more robust** than SL during pre-training for many cases, despite it producing more robust pre-trained models. In particular, we discover that CL really relies much more crucially than SL on the image spatial structure.
>
> * Explanation of the findings
>
> As mentioned in the submission, we claim our paper as an empirical study. While a thorough theoretical analysis is beyond the scope of this paper, we do hypothesize different reasons for the pre-training and the downstream robustness differences, and we have empirically validated these hypotheses.
>
> In the downstream, CL is more robust – We hypothesize the main reason is uniformity. We have measured the uniformity empirically in Sec. 5.1, and verified that CL indeed has higher uniformity than SL. We have also conducted an experiment showing increasing the feature uniformity during SL can improve SL’s downstream robustness.
>
> In the pre-training stage, CL is less robust than SL to pixel and patch-level corruptions. We hypothesize this may be due to the higher dependency of CL’s data augmentation strategy on the image structure in Sec. 5.2. To verify this, we show that the hypothetical setting of switching the order of augmentation and corruption significantly reduces the accuracy drop of CL.
>
> To sum up, we believe that our new findings based on comprehensive evaluations are of sufficient interest to the field. We hope they can inspire future work to study similar problems (e.g., behavioral difference of learning algorithms), provide comprehensive explanations to the discovered phenomenons, and ultimately work towards fully understanding CL.

---

> ### Author Response · Authors · 2022-11-21
> **Response to Reviewer XV1R [Part 2]**
>
> #### 2. [Unjustified metric]
>
> Thanks for bringing up the concern. The major reason for adopting relative accuracy drop (RAP) is the performance difference between CL and SL baselines (uncorrupted, original performance) on a large variation of datasets. For a simpler dataset where CL and SL baselines are close, RAP and absolute accuracy drop (AAP) will not have large differences in numbers; for a more difficult dataset where there is a large gap between CL and SL, AAP is inappropriate.
>
> Consider an extreme example. The baseline A = 90% accuracy, the baseline B = 20% accuracy, the same 10% AAP drop should matter more to the worse baseline because it is already very bad, but the baseline A dropping from 90% to 80% is still acceptable. This can be captured by the RAP but not by the AAP, since RAP(A) = 10/90 = 11% and RAP(B) = 10 / 20 = 50%, while AAP(A) = AAP(B) = 10%.
>
> To deal with the variations of baseline performance as well as the varying scale of performance drop on different datasets, we therefore measure the relative accuracy drop of each algorithm (as a normalization procedure) to minimize the impact of baseline gaps.
>
> On the other hand, we align the baseline performance of CL and SL algorithms for both pre-training and downstream and close the gap as much as possible.
>
> In fact, RAP is used in prior work to study robustness as well, such as Liu et al. ICLR 2022 [A] cited in our work that studies class imbalance.
>
> [A] Hong Liu, Jeff Z HaoChen, Adrien Gaidon, and Tengyu Ma. Self-supervised learning is more robust to dataset imbalance. In ICLR, 2022.
>
>
> #### 3. [Artificial corruption]
>
> Thanks for pointing this out. The goal of the artificial corruption with patch-level shuffling is to study what kind of image information is crucial for the pre-training algorithm. We briefly mention in Sec. 3.2 that we want to know **how CL and SL behave when we destroy the natural statistics of images manually**. This also motivates us to investigate the robustness behaviors of CL and SL during the pre-training stage. Importantly, as mentioned in the general response above, **the investigation on the patch-level corruption during pre-training leads to our novel discovery that CL really relies much more crucially than SL on the image spatial structure**.
>
>
> In fact, patch shuffling is the only artificial corruption out of the wide range of corruptions we have investigated that does not naturally occur in the real world. However, it is actually known in the literature. For example, Ge et al. [B] apply patch shuffling to negative samples to facilitate self-supervised pre-training. Note that, as mentioned in the general response above, most of the corruption types investigated in our paper are natural.
>
> [B] Songwei Ge, Shlok Mishra, Chun-Liang Li, Haohan Wang, and David Jacobs. Robust contrastive learning using negative samples with diminished semantics. In NeurIPS, 2021.

---

> ### Comment · Reviewer_XV1R · 2022-12-05
> **Thanks for your response**
>
> Thanks for your explanation. Now I understand why the author uses RAP as a metric and studies artificial corruption with patch-level shuffling. I also thank the author summarize the novelty/contribution of the submission in the rebuttal.
>
> After reading other reviewers' reviews and the author's response, my major concern is that the motivation of the setting studied and the implication of the findings are not clear enough. Thus I am afraid I cannot recommend the acceptance of the current manuscript of the work.
>
> However, I do believe the work has great potential to be beneficial if the motivation and implications of the findings can be articulated more clearly. The followings are my detailed feedback on the rebuttal.
>
> **New settings**
>
> Thanks for your explanation. I understand the work studies a new 'robustness' setting, **pre-training data corruption**. It would be good if the author can articulate why this setting is important and need to be studied.
>
> **New findings**
>
> *(a) While previous works reveal better robustness of CL than SL against out-of-domain and class distribution shifts, such an observation is limited to a few settings. We significantly enrich this observation to a wider set of corruption types and models, covering a hierarchy of corruption levels and 7 different contrastive learning algorithms.*
>
> It would be good if the author make arguments about why such an enriched observation of the better robustness of CL than SL is important. Based on your observation, can you provide more guidance to ML practitioners?
>
> *(b) Under-explored in the community and to the best of our knowledge, we are the first to observe that CL is not more robust than SL during pre-training for many cases, despite it producing more robust pre-trained models. In particular, we discover that CL really relies much more crucially than SL on the image spatial structure.*
>
> Similarly, it would be nice if the author can further explain how understanding CL being less robust during pre-training can inspire further improvements in the usage of CL and SL.

---

> > ### Author Response · Authors · 2022-12-10
> > **Response to follow-up questions [Part 1]**
> >
> > We thank you for your genuine feedback, and would like to further clarify your concerns.
> >
> > **1. [Why the pre-training corruption setting is important]**
> >
> > As stated in the previous response, our goal is to study how well pre-training algorithms work under pre-training data distributional changes. We feel existing research in the field overwhelmingly focuses on the **accuracy** of **natural, curated pre-training data** by developing new algorithms. The data side of things is less studied (until very recently). That is where our study fits in. We would like to further clarify from the following two aspects.
> >
> > **(1) Studying the pre-training corruption setting helps us to understand CL algorithms: what data information or image structure is critical for the high performance of CL algorithms**. By default, most of the existing CL algorithms are pre-trained on natural, curated, class-balanced data like ImageNet. Given that CL algorithms have no access to class semantic labels but they still learn good feature representations, a fundamental question arises: why is this the case? Our insight then is that the superior performance of CL algorithms might be due to the fact that they implicitly exploit certain image structures or information inherent in the existing datasets as inductive bias. What if we break the clean data assumption? This leads to our empirical investigation towards answering the question.
> >
> > To this end, we need to purposefully corrupt the pre-training data to deviate from the conventional assumption like natural images and class-balanced distribution and even produce unnatural images that do not occur in the existing curated datasets. Specifically, we systematically design the types of corruptions that destroy different aspects of image information: pixel-level – pixel intensity statistics, patch-level – image spatial structure, dataset-level – overall instance distribution. Our study shows that for several kinds of data corruption (e.g., pixel-level corrupted, patch-shuffled images), CL works quite poorly. This further **indicates that CL algorithms potentially highly rely on, for example, strong natural image spatial structure as inductive bias for achieving good performance, compared with supervised learning; once such spatial structure does not hold due to patch-level shuffling, their performance drops significantly**.
> >
> > **(2) Studying the pre-training corruption setting makes us aware of potential failure of existing CL algorithms and when**: CL algorithms are not as general as we might have thought; instead, they are actually customized to ImageNet-like, natural, object-centric images. In practice, there is often little control over pre-training data. This situation is akin to adversarial attacks, which also introduce natural/unnatural perturbations to the real data. Newly emerging models scrape images of mixed quality from the Internet (e.g., CLIP [1]), and more unsecured data will be adopted for pre-training, which is potentially corrupted by various kinds of corruption attacks. Therefore, our study raises valid concerns over such emerging issues in pre-training; and indeed, our study reveals the significantly decreased performance of CL algorithms with corrupted data.
> >
> > [1] Alec Radford, et al. Learning transferable visual models from natural language supervision. In ICML, 2021.

---

> > ### Author Response · Authors · 2022-12-10
> > **Response to follow-up questions [Part 2]**
> >
> > **2.a. [Why the enriched observation of downstream robustness of CL is important]**
> >
> > We would like to further clarify the importance of our enriched observation of the downstream robustness of CL from the following three aspects.
> >
> > * Our work enriches our understanding of CL with a significantly broader scope of downstream robustness. Our downstream robustness is different from prior works. Previous works focus on other aspects of downstream robustness as we discussed in the related work, such as adversarial and out-of-domain datasets. They work with **uncorrupted data** as opposed to **corrupted data** in our case. It was not obvious that the past findings could carry over without doing the experiments. Our experiments with diverse corruptions have filled this gap. We find CL is also more robust than SL to corruptions of downstream datasets.
> >
> > * Our work also provides a novel explanation about why CL is more robust than SL in downstream corruption. We hypothesize the more uniform feature space of CL directly **relates to** the more downstream *corruption* robustness, and design the experiment in Sec. 5.1 Table 7 to verify our hypothesis.
> >
> > * Our work further contributes to a novel finding on the contrasting observation of CL **between pre-training and downstream robustness**, which was not previously known. CL is actually less robust to pre-training pixel/patch corruptions than SL.
> >
> > As for the guidance to practitioners, (1) our observation of downstream robustness of CL reassures the trending use of CL pre-trained models in downstream tasks (over SL), even when the data undergo strong corruption so that they deviate from the clean data assumption (which is our novel observation). CL tends to learn more robust features that transfer better to downstream datasets than SL. (2) Our study also provides **quantitative** robustness measures under various corruptions, which is important in practical applications. (3) Another direct consequence is our Table 7, which improves SL by adding an extra CL-inspired uniformity regularization. We showed that adding the regularization helps improve the downstream robustness of SL. We hope our work inspires more exploration in this direction to provide additional guidance to practitioners.
> >
> >
> > **2.b. [How understanding CL being less robust during pre-training can inspire further improvements]**
> >
> > This reveals a major drawback of current CL algorithms, and reveals what image information in the *data* is crucial for the success of CL. They are not as general as we might have thought; instead, they are actually customized to ImageNet-like, natural, object-centric images. Our corruptions destroy different aspects of image information: pixel-level – pixel intensity statistics, patch-level – image spatial structure, dataset-level – overall instance distribution. CL algorithms are more sensitive to distributional changes caused by pixel-level and patch-level corruptions, which suggests they rely on natural image pixel statistics and the image spatial structure more heavily than SL.
> >
> > As for inspiring further improvements, (1) the main message is that people should be more cautious about using CL blindly on their new pre-training data source, without checking whether the data is suitable. (2) This may inspire researchers to develop better CL/SSL algorithms by proposing, e.g., more general objective functions or data augmentation strategies, which are applicable to wider kinds of data distributions. One objective function relevant to the patch-level corruption could be **equivariance** under patch shuffling – explicitly enforce the representation to be stable when the image is broken down into patches and the patches are shuffled.
> >
> > We look forward to your feedback, and we will further revise our manuscript with our additional clarifications.

---

### Official Review · Reviewer_fxue · 2022-10-31

**Confidence:** 3
**Correctness:** 3
**Technical Novelty And Significance:** 2
**Empirical Novelty And Significance:** 2
**Recommendation:** 5

**Clarity, Quality, Novelty And Reproducibility:**

Overall, this paper is partially novel in that the authors design a systemic framework for evaluating the behaviors of SL and CL frameworks on different data corruption strategies. However, there is a lack of in-depth analysis/comparison on behaviors of various CL algorithms. I think the proposed method is reproducible.

**Strength And Weaknesses:**

Strength
- This paper proposed a systemic framework for evaluating the behaviors of SL and CL frameworks with various data corruption strageies.
- The authors presented a number of metrics for measuring and analyzing feature learning, including feature semantic fluctuation, figure unfiormlmity, and fetuar distance.
- The authors demonstrate that uniformity may be the key to the success of contrastive learning on downstream robustness.
- Overall this paper is well-organized and easy to follow.

Weakness
- In section 3.3, the authors mention utilizing the same data augmentaions  for all methods. It is somewhat confusing. Did you apply any augmentation from the original CL algorithms?
- Some essential components of CL algorithms, such as batch size and temperature parameters, are not being discussed or studied. For instance, adjusting the batch size or temperature, for instance, could have a significant impact on the feature learning of negative samples for certain algorithms. Will this modification affect the current outcomes?
- It would be interesting to compare qualitatively how different corruption strategies impact the model's ability to learn semantic concepts/objects. Figure C.1 (attention maps) is a good illustration of how SL is superior to CL algorithms in capturing the main object of the image by applying patch shuffling.
- This paper lacks a comprehensive analysis or discussion of the distinctions between CL algorithms. In section 4.1, for instance, the authors note that CL algorithms perform differently on pixel-level corruption. Besides the numerical difference, are there any additional explanations for why this happened?
- In addition, I would suggest adding an average number of all datasets for each algorithm in Table 2 or Table B.5. This would make it easier for readers to compare the performance of different algorithms across all datasets.
- The authors added a uniformity-promoting loss term to supervise learning algorithm in table 7 of section 5.1. How do the authors define the term uniformity-promoting loss, for instance, the definition in 2 feature uniformity?


**Summary Of The Paper:**

This paper investigates the behavior of contrastive learning (CL) and supervised learning (SL) frameworks in response to changes in data distribution. To achieve this, they develop a variety of data corruption strategies, including patch, pixel, and dataset-level corruption. For experiments, they conduct a comprehensive comparison with a list of recent SSL algorithms. The authors further conclude that CL is more robust to dataset-level corruption than SL, but much less so to pixel- and patch-level corruption. In addition, the authors note that uniformity is the key to the downstream robustness of CL. Overall, this work offers a number of intriguing insights that could inspire future research in contrastive self-supervised learning.

**Summary Of The Review:**

Please check my comments in Strength And Weaknesses.

---

> ### Author Response · Authors · 2022-11-21
> **Response to Reviewer fxue [Part 1]**
>
> We sincerely thank you for your valuable suggestions. Below we clarify all of your concerns.
>
> #### 1. [General clarification on novelty]
>
> Before we address the specific concerns, we would like to first clarify our novelty, in the following aspects:
>
> * New settings
>
> We want to emphasize that, in addition to the downstream data corruption, our study also investigates **pre-training data corruption**.
>
> The two settings characterize two complementary aspects of distributional robustness. The downstream corruption studies how a pre-trained representation space copes with distribution-shifted data. The pre-training corruption studies how the learning algorithm performs with distribution-shifted training data.
>
> Our settings are novel, especially the pre-training corruption setting. As noted by Reviewer J9U1 and Reviewer CG2S, most distributional robustness papers only focus on the *downstream* setting with *uncorrupted* data. A systematic comparison of SL and CL to a wide range of *pre-training* corruptions is missing in the literature.
>
> * Comprehensive corruption types
>
> Our data corruptions contain pixel-level (manipulating pixel values), patch-level (destroying global or local structural information), and dataset-level (changing dataset distribution), for us to comprehensively evaluate the impact of corrupted data in the downstream, and how such corruptions interfere with feature learning. Such comprehensiveness does not exist in prior works, thus making our conclusions more general and well-supported.
>
> Importantly, we consider **both natural and unnatural** corruptions. Most of the corruption types are natural: shot-noise, blur, compression, gamma distortion, as well as class imbalance and synthetic data, and they can all happen in real life. The patch shuffling corruption is unnatural, but we specifically choose this corruption to study whether breaking down the image spatial structural information affects feature learning.
>
> * New findings
>
> 1) While previous works reveal better robustness of CL than SL against out-of-domain and class distribution shifts, such an observation is limited to a few settings. We significantly enrich this observation to a wider set of corruption types and models, covering a hierarchy of corruption levels and 7 different contrastive learning algorithms.
>
> 2) Under-explored in the community and to the best of our knowledge, we are the first to observe that CL is **not more robust** than SL during pre-training for many cases, despite it producing more robust pre-trained models. In particular, we discover that CL really relies much more crucially than SL on the image spatial structure.
>
> * Explanation of the findings
>
> As mentioned in the submission, we claim our paper as an empirical study. While a thorough theoretical analysis is beyond the scope of this paper, we do hypothesize different reasons for the pre-training and the downstream robustness differences, and we have empirically validated these hypotheses.
>
> In the downstream, CL is more robust – We hypothesize the main reason is uniformity. We have measured the uniformity empirically in Sec. 5.1, and verified that CL indeed has higher uniformity than SL. We have also conducted an experiment showing increasing the feature uniformity during SL can improve SL’s downstream robustness.
>
> In the pre-training stage, CL is less robust than SL to pixel and patch-level corruptions. We hypothesize this may be due to the higher dependency of CL’s data augmentation strategy on the image structure in Sec. 5.2. To verify this, we show that the hypothetical setting of switching the order of augmentation and corruption significantly reduces the accuracy drop of CL.
>
> To sum up, we believe that our new findings based on comprehensive evaluations are of sufficient interest to the field. We hope they can inspire future work to study similar problems (e.g., behavioral difference of learning algorithms), provide comprehensive explanations to the discovered phenomenons, and ultimately work towards fully understanding CL.
>
> #### 2. [Data augmentation]
>
> Yes, we apply the **standard** data augmentations from the CL algorithms. We also unify the order of data augmentations for all CL and SL algorithms in our experiments, with appropriate hyper-parameters corresponding to each dataset, so that we have a fair comparison between methods on each dataset. Specifically, we include standard augmentations in the following order: RandomResizeCrop, RandomApply(ColorJitter), RandomGrayscale, RandomApply(GaussianBlur), and RandomHorizontalFlip. Please refer to Table A.1 in the Appendix for more details.

---

> ### Author Response · Authors · 2022-11-21
> **Response to Reviewer fxue [Part 2]**
>
> #### 3. [CL hyper-parameters]
>
> The change of hyper-parameter settings will not affect our current outcomes. This is because our investigation on robustness depends on the **relative performance change** of a CL algorithm when the data change from uncorrupted to corrupted, rather than its **absolute performance**. Moreover, our investigation focuses on the comparison between the robustness of SL and CL algorithms. For this purpose, we have tuned the hyper-parameters of CL algorithms to obtain uncorrupted accuracy that is as high as being comparable with the uncorrupted SL accuracy; and then we conduct CL experiments under data corruptions with the same hyper-parameters. In our  experiments, we found that when we use a different set of hyper-parameters, for CL algorithms it will lead to both lower uncorrupted and corrupted accuracies than our current choice of hyper-parameters, but our conclusions on the robustness comparison between SL and CL algorithms still hold.
>
> #### 4. [Qualitative comparison of learning semantics]
>
> Per the reviewer’s request, in Appendix C.1 of the revision, to further understand qualitatively how different corruption strategies impact the model’s ability to learn semantic concepts, we draw the class activation maps (CAMs) of models trained under different corruption settings on the corrupted versions of two ImageNet validation images in Figures C.2 and C.3. Consistent with the observation in the original Figure C.1, MoCo is worse than SL in capturing the main object of the image. Figures C.2 and C.3 further show that global shuffling and defocus blur especially hinder the ability of MoCo to learn meaningful semantics.
>
> #### 5. [Distinctions between CL algorithms]
>
> We thank the reviewer for the question. As the paper’s title indicates, we primarily focus on the difference between CL and SL algorithms – we believe this is already of significance. And the important distinction we discovered in this paper is between CL and SL algorithms. Our conclusions that 1) CL is more robust to downstream data corruptions than SL and 2) CL is sensitive to pre-training pixel/patch-level corruptions are generally true for all CL algorithms which we have investigated.
>
> In fact, the reviewer’s question further reiterates the importance of our work – the behavioral differences of learning algorithms are significantly overlooked in the community, and there are a lot of open questions to be answered. As we discussed in Sec. 3.3, various CL algorithms stem from different theoretical bases and approaches (e.g., with or without negatives, clustering, redundancy reduction, etc.) and may have fine-grained behavioral differences. Providing a more in-depth explanation to this question is beyond the scope of our paper, and we leave the study of fine-grained distinctions among CL algorithms as interesting future work.
>
> #### 6. [Average number of all datasets]
>
> Thanks for the suggestion. We have updated both tables with average across all datasets for each algorithm in the manuscript.
>
> #### 7. [Uniformity-promoting loss]
>
> We apologize for the confusion. Yes, you are right. The uniformity-promoting loss term is simply the negative of definition 2 feature uniformity, computed on a mini-batch multiplying a loss coefficient 0.01.

---

> ### Author Response · Authors · 2022-12-10
> **Update: Additional experiments on CL hyper-parameters**
>
> Again, we thank the reviewer for the suggestion. As mentioned in the previous response, the change of hyper-parameter settings will not affect our current outcomes. Here we provide additional experimental validation in the following table, which tunes the hyper-parameters of MoCo-v2 on each corrupted CIFAR10 setting as in Table 3. For a fair comparison, we also provide the result of tuning the hyper-parameters of supervised learning on each corrupted setting. We searched various sets of hyper-parameters over a wide range, including CL-specific terms such as temperature. **The improvements with tuning hyper-parameters are marginal, and our conclusions remain the same as in the submission**. We will include these results in the revision.
>
> | Model | gam 0.2 | glo 4 | glo 8 | loc 4 | loc 8 | avg delta |
> |:---:|:---:|:---:|:---:|:---:|:---:|:---:|
> | sup (89.53) submission | 87.36 (2.4%) | 76.06 (15.0%) | 65.88 (26.4%) | 65.94 (26.3%) | 77.49 (13.4%) | 16.7% |
> | sup-tuned | 87.46 (2.3%) | 76.90 (14.1%) | 66.24 (26.0%) | 66.43 (25.8%) | 77.93 (13.0%) | 16.2% |
> | MoCo-v2 (88.73) submission | 85.84 (3.3%) | 67.18 (24.3%) | 60.51 (31.8%) | 63.35 (28.6%) | 76.90 (13.3%) | 20.3% |
> | MoCo-v2-tuned | 86.17 (2.9%) | 67.92 (23.5%) | 62.70 (29.3%) | 63.81 (28.1%) | 77.07 (13.1%) | 19.4% |
>
> We look forward to your feedback.

---

### Author Response · Authors · 2022-11-21
**General response to all reviewers**

Dear reviewers,

We sincerely thank you for your valuable comments. We have revised our submission (highlighted in red) to address 1) explicit clarifications of terminologies (p.4), 2) additional citation (p.3), 3) additional CAM visualizations (p.20,21), 4) table updates with more statistics (p.6,17), and 4) additional result measuring task difficulty by h-divergence (p.18). We hope the revision can better demonstrate our observations and clarify the confusions you have. We are looking forward to your further feedback.

---

### Author Response · Authors · 2022-11-28
**General reminder to all reviewers**

Dear reviewers,

It is a friendly reminder regarding the manuscript revision and responses to your reviews. We thank you again for your valuable comments, and look forward to further discussions and feedback.

---

### Decision · Program_Chairs · 2023-01-20

**Decision:**

Reject

**Justification For Why Not Higher Score:**

Motivation and implications unclear.

**Justification For Why Not Lower Score:**

n/a

**Metareview: Summary, Strengths And Weaknesses:**

In this work, the authors investigate the differences in behavior of learned representations from supervised and contrastive learning. In particular, the authors focus on the robustness of the representations to downstream tasks in the presence of pixel-level, patch-level and dataset-level corruptions. The authors find that representations learned through contrastive learning are more robust to dataset-level corruptions than supervised learning; this robustness however is much less prominent with pixel-level and patch-level corruptions. The authors explore these questions on ResNet-50 and ViT architectures and test on ImageNet, CIFAR-10/100 and a litany of other small image recognition tasks. Additionally, the authors test a range of contrastive learning approaches including MOCO, BYOL, SimCLR, etc.

The reviewers commented positively on the systematic framework for evaluating the behaviors of supervised and contrastive learning, the organization and presentation of the material and numerous experiments conducted by the authors. The reviewers also expressed some concern about the lack of in-depth analysis of behaviors of various critical contrastive learning parameters (e.g. temperature, batch size, etc.); the lack of motivation about some of the corruption studies, particularly pertaining to pretraining, since many of the corruptions appear rather unnatural; questions about the overall implications of the study. With regard to the motivation, there was some discussion between the reviewers and authors but the reviewers were unconvinced particularly in the setting of corruptions applied to data during pretraining. As to the overall implications of the study, the results of this study were unclear. There were previous studies that have already explored how the performance of downstream tasks changes in the context of corruption, thus the novelty for similar results reported in this paper are limited. Likewise, given the concerns about motivation on corruptions applied during pretraining, it was difficult to state the new implications provided by this study.

Given that none of the reviewers felt that their concerns were addressed, this paper will not be accepted at this conference. From my perspective, I would encourage the authors to revise the motivation for their study, particularly in light of the previous literature, and consider resubmitting to a future venue.